# Neural dynamics differentially encode phrases and sentences during spoken language comprehension

**Fan Bai**[1,2], **Antje S. Meyer**[1,2], **Andrea E. Martin**[1,2]*

**1** Max Planck Institute for Psycholinguistics, Nijmegen, the Netherlands, **2** Donders Institute for Brain, Cognition, and Behaviour, Radboud University, Nijmegen, the Netherlands

* andrea.martin@mpi.nl

**Data Availability Statement:** All data and analysis files are available from the Max Planck Institute for Psycholinguistics data repository https://hdl.handle.net/1839/dc743f61-65fc-4b6b-979f-

## Abstract

Human language stands out in the natural world as a biological signal that uses a structured system to combine the meanings of small linguistic units (e.g., words) into larger constituents (e.g., phrases and sentences). However, the physical dynamics of speech (or sign) do not stand in a one-to-one relationship with the meanings listeners perceive. Instead, listeners infer meaning based on their knowledge of the language. The neural readouts of the perceptual and cognitive processes underlying these inferences are still poorly understood. In the present study, we used scalp electroencephalography (EEG) to compare the neural response to phrases (e.g., the red vase) and sentences (e.g., the vase is red), which were close in semantic meaning and had been synthesized to be physically indistinguishable. Differences in structure were well captured in the reorganization of neural phase responses in delta (approximately <2 Hz) and theta bands (approximately 2 to 7 Hz),and in power and power connectivity changes in the alpha band (approximately 7.5 to 13.5 Hz). Consistent with predictions from a computational model, sentences showed more power, more power connectivity, and more phase synchronization than phrases did. Theta–gamma phase–amplitude coupling occurred, but did not differ between the syntactic structures. Spectral–temporal response function (STRF) modeling revealed different encoding states for phrases and sentences, over and above the acoustically driven neural response. Our findings provide a comprehensive description of how the brain encodes and separates linguistic structures in the dynamics of neural responses. They imply that phase synchronization and strength of connectivity are readouts for the constituent structure of language. The results provide a novel basis for future neurophysiological research on linguistic structure representation in the brain, and, together with our simulations, support time-based binding as a mechanism of structure encoding in neural dynamics.

## Introduction

Speech can be described by an abundance of acoustic features in both the time and frequency domains [1–3]. While these features are crucial for speech comprehension, they do not

fdd0791beda4 The underlying data can be found in
https://doi.org/10.5281/zenodo.6595789.

**Funding:** AEM was supported a Max Planck Research Group and a Lise Meitner Research Group "Language and Computation in Neural Systems" from the Max Planck Society, and by the Netherlands Organization for Scientific Research (NWO; grant 016.Vidi.188.029). The funders had no role in study design, data collection and analysis, decision to publish, or preparation of the manuscript.

**Competing interests:** The authors have declared that no competing interests exist.

**Abbreviations:** DFT, discrete Fourier transform; DORA, Discovery of Relations by Analogy; EEG, electroencephalography; FIR, finite impulse response; HDI, highest density interval; ICA, independent component analysis; ISPC, intersite phase coherence; ITPC, intertrial phrase coherence; MCMC, Markov Chain Monte Carlo; MEG, magnetoencephalography; PAC, phase amplitude coupling; RMS, root mean square; ROI, region of interest; ROPE, region of practical equivalence; RSM, representational similarity matrix; SF, stimulus feature; SNR, signal to noise ratio; SRF, spectral response function; STRF, spectral temporal response function; STFT, short-time Fourier transform; TRF, temporal response function.

themselves signpost the linguistic units and structures that give rise to meaning, which is, in turn, highly determined by morphemic and syntactic structure. Spoken language comprehension therefore relies on listeners going beyond the information given and inferring the presence of linguistic structure based on their knowledge of language. As such, many theories posit that linguistic structures—ranging from syllables to morphemes to "words" to syntactic structures—are constructed via an endogenous inference process [4–19]. On this view, also known as "analysis by synthesis" [20], speech triggers internal generation of memory representations (synthesis), which are compared to the sensory input (analysis). This results in linguistic structures that come about through the synthesis of internal brain states (viz, linguistic knowledge) with sensory representations via perceptual inference [9,10,17,21–23]. Recent studies have begun to investigate the neural activity that corresponds to the emergence of linguistic structure [24–28], in particular in terms of temporal and spatial dynamics of brain rhythms. However, despite these efforts, details of how the brain encodes or distinguishes syntactic structures remain unknown. The main question we address here is which neural readouts (or measurements of the neural response) are relevant for tracking the transformation of a physically and temporally near-identical (viz., statistically indistinguishable) speech input into 2 different abstract structures, in this case, a phrase and a sentence. This transformation process is interesting because it is an instance of brain computation where the brain takes a physical stimulus that would seem identical to another species or machine and computes different structural properties based on culturally acquired abstract knowledge. In order to understand this neural computation better, an important first step (in addition to theoretical and computational modeling) is to know which neural readouts are relevant, in order to be able to use those readouts in the future to constrain how we build our theories and models. As such, in this study, we investigated the neural responses to minimally different linguistic structures, specifically phrases like "the red vase" compared to sentences like "the vase is red." We investigated which dimensions of neural activity discriminated between the linguistic structure of these phrases and sentences, by minimizing their differences in acoustic–energetic/temporal–spectral profiles and semantic components, such that these differences became statistically indistinguishable (see Methods for acoustic normalization and analysis).

## Low-frequency neural oscillations and linguistic structure

A growing neuroscience literature suggests that low-frequency neural oscillations (approximately <8 Hz) are involved in processing linguistic structures [11,12,24–27,29–38]. In a highly influential study by Ding and colleagues[24], low-frequency power "tagged" the occurrence of phrases and sentences in well-controlled speech stimuli. That is, power increases were observed that coincided with occurrences of phrases or sentences at a fixed rate of 2 and 1 Hz, respectively. Additionally, the grouping of words into phrases of different lengths modulated the location of the frequency tag accordingly, indicating that power at certain frequencies could track linguistic structures. Subsequent research has further confirmed the sensitivity of oscillatory power and phase in the delta band (approximately 2 to 4 Hz) to higher level linguistic structures like phrases [25,27,29,39,40].

Inspired by these empirical studies, Martin and Doumas [11] proposed a computationally explicit framework for modeling the role of low-frequency neural oscillations in generating linguistic structure (based on the symbolic-connectionist model of relational reasoning Discovery of Relations by Analogy (DORA), [41]). The core mechanism that the model proposed is *time-based binding*, where the relative timing of neural ensemble firing functions as a way to carry information forward in a neural system. As such, relative timing of firing operates as a degree of freedom for expressing relations between patterns in the system; distributed

representations that fire together closely in time pass activation forward to "receiver" ensembles. For example, if the distributed pattern of activation for a word occurs closely in time to that of another word, and the words can be combined into a phrase, a downstream ensemble functions as a receiver, sensitive to the occurrence of compatible patterns occurring together in time, and thus forming larger units like phrases and sentences. In this framework, which was extended into a theoretical model [10], the authors reproduced the frequency tagging results reported by Ding and colleagues [24] in an artificial neural network model that used time (viz., asynchrony of unit firing) to encode structural relations between words (see also [12]). The model exploited asynchrony of unit firing to form structures from words; the temporal proximity of unit firing was used to encode linguistic structures and relations related to the word-level input. More specifically, nodes on a higher layer (i.e., localist units representing phrasal units) code for the composition of sublayer units (i.e., words) in this network structure. The higher-level node fires when any of its sublayer nodes fire in time, forming a "phase set" between words and their phrase. In this implementation, linguistic structures were represented neural dynamics across a layered network, where the asynchrony of unit firing (i.e., firing staggered in time) not only allowed the network to combine lower-level representations together for processing on a higher-level, but also served to keep the lower-level representations independent as well [11,12,41]. Based on this model, Martin and Doumas hypothesized that activation in the network should depend on the number of constituents that are represented at a given timestep; namely, the more phrase or structural relations that are being computed, the more units are active, resulting in more power and connectivity in the network. Because units must fire together in time in a systematic pattern in order to represent a structure, there will be more phase synchronization as a function of the constituents or structural units that are being represented. In the model's representational coding scheme, constituents are represented as (localist) relations between distributed representations in time. Thus, the ongoing dynamics of neural ensembles involved in coding linguistic units and their structural relations are what constitute "linguistic structure" in such a neural system [9,10]. The current study tested this prediction using Dutch phrases (e.g., *de rode vas*, "the red vase") and sentences (e.g., *de vas is rood*, "the vase is red") that were closely matched in acoustic-physical characteristics, semantic features, and interpretation, but differed in morphemic and syntactic structure (viz., inflectional morphemes, the number and type of constituents perceived; see Methods). In order to confirm and illustrate the predictions for the current study, we performed simulations of time-based binding and DORA's activation in response to our stimuli and then analyzed the activation in the model using neural time series methods (for details, see Results and Methods).

## Low-frequency oscillations and speech intelligibility

Low-frequency neural activity, in particular phase coherence in the approximately 2 to 7 Hz theta band, is highly correlated with spoken language comprehension [42–45]. For example, a magnetoencephalography (MEG) study by Luo and Poeppel [44] showed that theta-band phase coherence was positively correlated with speech intelligibility. One possible explanation of such findings is that people's ability to infer linguistic structure increases with speech intelligibility, with resulting changes in low-frequency neural activity (see also [24]).

Low-frequency neural oscillations may be especially important for speech processing because they occur roughly at the average syllable rate across human languages [46–48]. The brain may use syllables, which are abstract linguistic units, as the primitive units to analyze spoken language [44,49,50]. Indeed, a view has emerged wherein the brain employs an inherent cortical rhythm at a syllabic rate that can be altered by manipulations of linguistic structure or intelligibility. One possible synthesis of previous results is that low-frequency power reflects

the construction of linguistic structures [24–26], whereas low-frequency phase coherence reflects parsing and segmenting of speech signals [42–45]; though we note that the relationship between power and phase in the neural signal is complicated, if not fraught. However, the format in which the brain represents higher-level linguistic structures remains unknown, and novel insights into this issue could have substantial implications for theories of speech and language comprehension.

## The current study

We investigated whether low-frequency neural oscillations reflect differences in syntactic structure. In order to increase the likelihood that any observed patterns are due to representing and processing syntactic structure, we strictly controlled the physical and semantic features of our materials. We extend the work of Ding and colleagues [24] and others to ask whether the 1 Hz neural response can be decomposed to reflect separate syntactic structures (phrases versus sentences). To assess this, we used 2 types of natural speech stimuli in Dutch, namely determiner phrases such as *De rode vaas* (The red vase) and sentences such as *De vaas is rood* (The vase is red), which combines the determiner phrase *De vaas* with the verb *is* and the adjective *rood* to form a sentence. Phrases and sentences were matched in the number of syllables (4 syllables), the semantic components (same color and object), the duration in time (1 second, sampling rate 44.1 k Hz), and the overall energy (root mean squared value equals −16 dB). We note that in order to achieve these syntactic differences morphemic differences are necessary. The inclusion of the verb *is* renders *De vaas is rood* a sentence, while the presence of the inflectional agreement morpheme *-e* renders *De rode vaas* a phrase; because the word *rood* modifies *vaas* in the phrase condition, it must bear a common gender agreement morpheme (*-e*) in order to be grammatical in Dutch (see Methods for more detail). The long vowel denoted by *oo* in the uninflected *rood* is the same vowel as in *rode*, but due to orthographic convention, it is written as a single *o* when the presence of the morpheme *-e* results in an open syllable at the end of the word. Morphemic differences (viz., an inflectional morpheme *-e* on the adjective *rood* in the Phrase condition, and verb/ inflectional verb phrase headed by *is* in the Sentence condition) cue the construction of hierarchy (and of syntactic structure more generally). In our stimuli, we were able to introduce these features while making the speech stimulus energetically and spectrotemporally indistinguishable from a statistical point of view. The necessity of the morphemes to form the different syntactic structures does indicate, however, that it is likely impossible to fully orthogonalize morphemic and syntactic information in languages like Dutch and English.

We formulated a general hypothesis that low-frequency neural oscillations would be sensitive to the difference in syntactic structure between phrases and sentences. However, we did not limit our analysis to low-frequency power and phase, as in previous research [24–26,29]. We hypothesized that the neural response difference between phrases and sentences may manifest itself in a number of dimensions, dimensions that are outside of the view of typical analyses of low-frequency power and phase.

We therefore employed additional methods to decompose the neural response to phrases and sentences, to address the following 5 questions:

**Question 1:** Do phrases and sentences have different effects on brain dynamics as reflected at the functional neural network level (viz., functional connectivity); specifically, do sentences result in more connectivity or phase synchronization as a function of structural units? Neuroscience has exhibited a rapidly growing interest in investigating functional connectivity in order to study whole brain dynamics in sensor space [51–55], which can reveal temporal synchronization (viz., phase coherence) between brain regions. Neurophysiological techniques

such as EEG and MEG have a high temporal resolution and are suitable for calculating synchronization across frequency bands in functional brain networks [56]. Describing the temporal synchronization of the neural activity over the whole brain is the first step in decomposing neural responses to high-level variables like syntactic structure. We therefore investigated whether phrases and sentences have different effects on intertrial phrase coherence (ITPC) and intersite phase coherence (ISPC), which are considered to reflect the temporal synchronization of neural activity [53,57,58].

**Question 2:** Do phrases and sentences differ in the intensity with which they engage connected brain regions? Power connectivity [52,53,59] can be used to describe a functional neural network in terms of the energy that is expended during the performance of a cognitive task. Power connectivity is a measure of how different underlying brain regions are connected via the intensity of induced neural responses in the time–frequency space. Differences in power connectivity would imply that phrases and sentences differentially impact the distribution and intensity of neural networks involved in speech and language comprehension. Therefore, we want to know whether the syntactic structure discrimination between phrases and sentences would be reflected in the neural activity of the organized network, and specifically whether sentences incur more power connectivity than phrases.

**Question 3:** Do phrases and sentences have different effects on the coupling between lower and higher frequency activity? This question is related to Giraud and Poeppel's theoretical model of a generalized neural mechanism for speech perception [49]. The model, which is focused on syllable-level processing, suggests that presentation of the speech stimulus first entrains an inherent neural response at low frequencies (less than 8 Hz) in order to track to the speech envelope, from which the neural representation of syllables is then constructed. Then, the low-frequency neural response evokes a neural response at a higher frequency (25 to 35 Hz), which reflect the brain's analysis of phoneme-level information. The model proposes coupling between low and high frequency neural responses (theta and gamma, respectively) as the fundamental neural mechanism for speech perception up to the syllable. We therefore investigated whether theta–gamma frequency coupling may also differentiate higher-level linguistic structure, namely phrases and sentences.

**Question 4:** Do phrases and sentences also differentially impact neural activity at higher frequencies such as the alpha band? The functional role of alpha-band oscillations in perception and memory is widely debated in systems neuroscience. However, whereas a role for low-frequency neural activity in language processing is beyond doubt, whether alpha-band activity has an important contribution has not yet been established. Alpha-band activity correlates with verbal working memory [60–63] and auditory attention [64–67]. Some oscillator models of speech perception consider the induced neural response at alpha as a "top-down" gating control signal [49,68], reflecting domain-general aspects of perceptual processing that are not specific to language processing. Other researchers argue that alpha-band activity reflects speech intelligibility [69–71], which opens up a potential role for alpha-band oscillations in syntactic or semantic processing. We therefore investigated whether phrases and sentences elicited differences in alpha-band activity.

**Question 5:** Can we obtain evidence for the differential encoding of phrases and sentences after "modeling out" physical differences? Neural responses to phrases and sentences comprise a mixture of processes associated with linguistic-structure building and with processing acoustic stimuli. To deal with this issue, one can model which aspects of the neural response encode the acoustic information, and then detect differences between phrases and sentences in the remainder of the neural activity, from which acoustic differences have been regressed out. Previous research using this approach, the spectral–temporal response function (STRF) shows that low-frequency neural responses robustly represent the acoustic features in the speech

[72–74] and that phoneme-level processing is reflected in the low-frequency entrainment to speech [75–77]. We therefore used the STRF to investigates which dimensions of the neural responses reflect differences between phrases and sentences.

In sum, we investigated different dimensions of the electroencephalography (EEG) response to spoken phrases and sentences. Observing differences between phrases and sentences would serve as a trail marker on the path towards a theory of the neural computations underlying syntactic structure formation, and lays the foundation for a theory of neural read-outs that are relevant for structure-building during language comprehension.

## Results

### Hypotheses based on a time-based binding mechanism for linguistic structure across neural ensembles

To test the predictions based on the time-based binding mechanism, we performed a simulation using the two different types of linguistic structure, namely phrases and sentences (for details, see Methods). The model's representations of the two syntactic structures are shown in **Fig 1A** and **1B**. Strong evidence for a link between time-based binding and observed data would be the ability to make fine-grained quantitative predictions about the magnitude of a coupling effect and its timing. However, there is a major impediment to that state of affairs—namely that precise timing predictions for coupling cannot be derived from any model without explicit knowledge about interregion connectivity. Another way to describe this constraint is that nodes in the model represent groups of neurons that could potentially reside at different locations in the brain (viz., DORA is a model of aggregate unit activity, where nodes represent population-level responses, not the responses of individual neurons). Given the varied properties of the intersite communication in both temporal and spectral dimensions [52,53,78], we cannot give a specific prediction regarding either the frequencies nor the positions (in the brain) of the effect, or of the precise temporal distribution of the effect, other than the time window in which it should occur given how linguistic structure is represented using time-based binding in DORA. In other words, we can simulate the boundary conditions on phase coupling given the representational and mechanistic assumptions made by time-based binding and DORA for representing and processing linguistic structure. The initial binding difference begins at approximately the second S-unit (PO-2 for both conditions), or the onset of the second syllable. We predict that the phase coherence difference should begin to occur within the range of approximately 250 ms to 500 ms. In addition, in line with previous studies, we predict that low-frequency neural oscillations will be highly involved in the discrimination of the two types of linguistic structure. To un-bias the estimation, all information for statistical inference was extracted in a frequency combined approach (<13.5 Hz). Based on the unfolding of syntactic structure in the computational model, we demonstrate that increased power and phase coherence occurs in the model during the time window of 250 to 1000 ms.

We first calculated the power activation using DORA's PO-units, where differences in binding between units initially occurs as the structures unfolded. The spectrally decomposed activity is shown in **Fig 1C**. Statistical analysis using paired-sample $t$-test on the frequency combined response suggested that power of activation was significantly higher for sentences than for phrases (t (99) = 8.40, $p$ < 3.25e-13, ***). The results are shown in **Fig 1D**. We then calculated the level of power coupling (connectivity) between PO-units and S-units for both conditions (**Fig 1E**). Statistical inference using paired sample $t$-test indicated that the level of power coupling was significantly higher for sentences than for phrases (t (99) = 251.02, $p$ < 1e-31, ***). The results are shown in **Fig 1F**. As we are also interested in the difference of phase alignment between the two conditions, we calculated phase coherence and phase coupling.

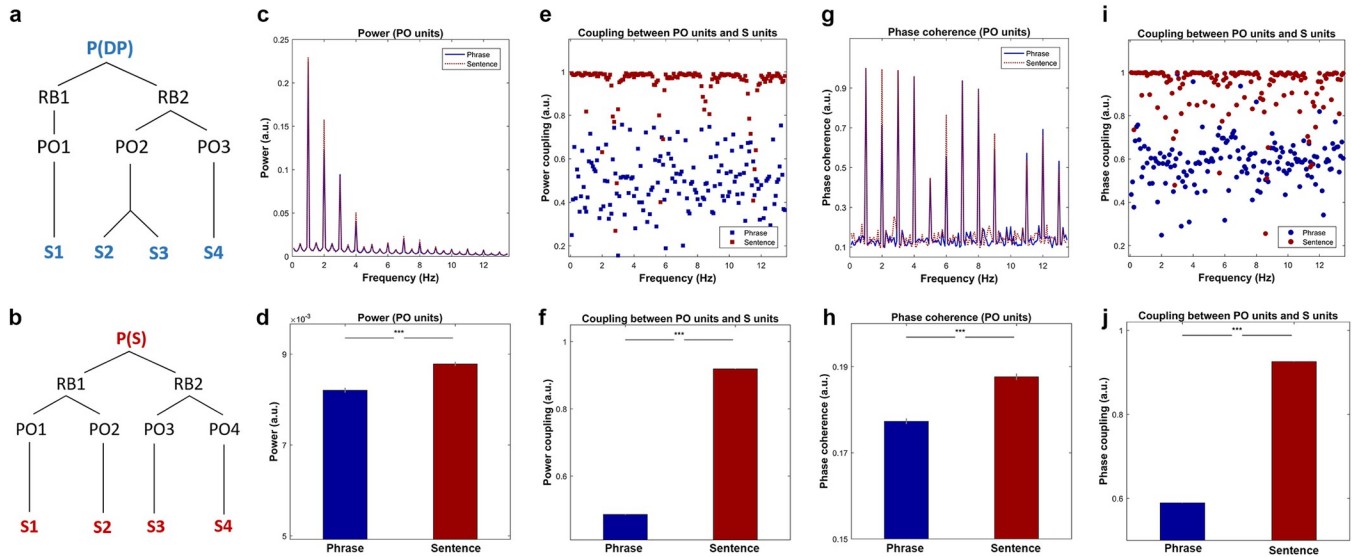

**Fig 1. Simulation results based on the time-based binding mechanism.** Simulation results based on the time-based binding hypothesis. **(a)** and **(b)** The model representation of phrases and sentences, in which the P (Proposition units), RB (Role-filler units), PO (Propositional Object units), and S (syllables units) represent the different types of node in DORA. P(DP) represents the top-level unit is a DP and P(s) represents the highest-level unit is a sentence. **(c)** Simulation results on power, in which the red dotted line and blue solid line represent the frequency response of the sentences and the phrases, respectively. The shading area covers 2 SEM centered on the mean. **(d)** Statistical comparison on the frequency combined power using paired sample *t* test suggested that the power for the sentences was significantly higher than the phrases (t (99) = 8.40, *p* < 3.25e-13, ***). **(e)** Results of power coupling between PO units and S units, where the red and blue squares show the frequency separated coupling level for the sentences and the phrases, respectively. **(f)** Statistical comparison on the level of power coupling using paired sample *t* test suggested that the power coupling level for the sentences was significantly higher than the phrases (t (99) = 251.02, *p* < 1e-31, ***). **(g)** Results of phase coherence, in which the red-dotted line and blue-solid line shows the phase coherence of the sentences and the phrases, respectively. The shading area represents 2 SEM centered on the mean. **(h)** Statistical comparison of the phase coherence was conducted using paired sample *t* test on the frequency averaged response. The comparison indicates that the phase coherence of the sentences was significantly higher than that of the phrases (t (99) = 10.24, *p* < 1e-20, ***). **(i)** Phase coupling between PO units and S units, the red and blue circles show the level of phase synchronization of the sentences and the phrases, respectively. **(j)**. Statistical comparison on the level of phase coupling between the phrases and the sentences using paired sample *t* test suggested that the phase coupling level for the sentences was significantly higher than the phrases (t (99) = 296.03, *p* < 1e-39, ***). DP, determiner phase.

**Fig 1G** shows the results of phase coherence on PO-units, statistical inference conducted using paired sample *t*-test on the averaged phase coherence suggested that the phase coherence for the sentences was significantly higher than the phrases (**Fig 1H**, t (99) = 10.24, *p* < 1e-20, ***). **Fig 1I** shows the results of phase coupling between PO-units and S-units, statistical comparison using a paired sample *t*-test on the frequency-averaged phase synchronization indicated that the level of phase synchronization was significantly higher for sentences than for phrases (**Fig 1J**, t (99) = 296.03, *p* < 1e-39, ***). In sum, based on the time-based binding mechanism, the results of the simulation demonstrated that the degree of phase synchronization between layers of nodes varies when different types of syntactic structure are processed. In addition, intersite connectivity for both power and phase should be higher for sentences than phrases. Due to the unknowable temporal and spatial aspects that are inherent in simulation (e.g., missing information as to where the neural ensembles modeled in the simulation occur in the brain and how this distribution then affects the measurement of coupling), we conducted our EEG experiment in order to observe whether empirical measurements of phase coupling can function as a neural readout that is sensitive to linguistic structure.

## Low-frequency phase coherence distinguishes phrases and sentences

To answer our first question, whether low-frequency neural oscillations distinguish phrases and sentences, we calculated phase coherence (for details, see Methods). We then performed a

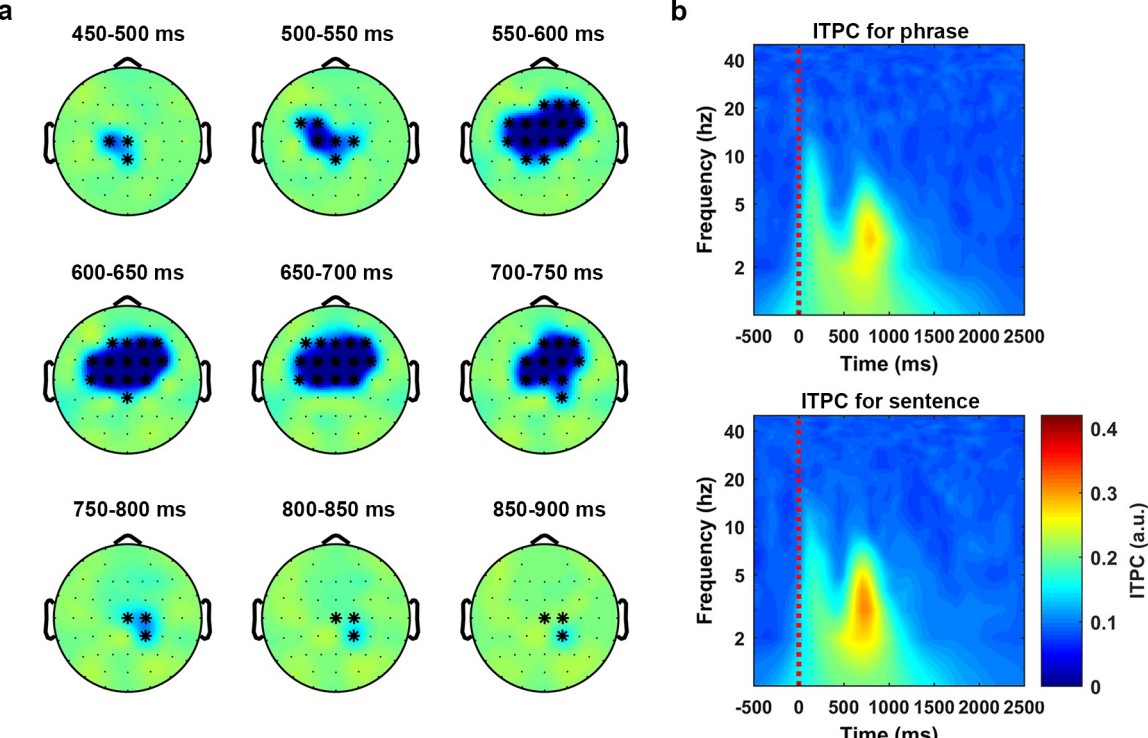

**Fig 2. Phase coherence separates phrases and sentences at theta band.** Statistical analysis on the phase coherence (ITPC) was conducted using a nonparametric cluster-based permutation test (1,000 times) on a 1,200-ms time window, which started at the audio onset and over the frequencies from 2 Hz to 8 Hz. The results indicated that the phase coherence was higher for the sentences than the phrases ($p < $ 1e-4 ***, 2-tailed). **(a)** The temporal evolution of the cluster that corresponds to the separation effect. The activity was drawn using the ITPC of the phrases minus the ITPC of the sentences. The topographies were plotted in steps of 50 ms. **(b)** ITPC averaged over all the sensors in the cluster. The upper panel and the lower panel show the ITPC of the phrases and the sentences, respectively. ITPC, intertrial phrase coherence.

nonparametric cluster-based permutation test (1,000 permutations) on a 1,200-ms time window starting at the audio onset and over the frequencies from 1 Hz to 8 Hz. The comparison indicated that phase coherence was significantly higher for sentences than phrases ($p < $ 1e-4 ***, 2-tailed). In the selected latency and frequency range, the effect was most pronounced at central electrodes.

Fig 2A shows the temporal evolution, in steps of 50 ms, of the discrimination effect which is computed as the averaged phase coherence of the phrases minus the averaged phase coherence of the sentences. Fig 2B shows the time–frequency decomposition using all the sensors in the cluster, in which the upper and lower panel are the plot for the phrase condition and the sentence condition, respectively.

The results indicated that the low-frequency phase coherence could reliably distinguish phrases and sentences, consistent with the hypothesis that low-frequency phase coherence represents cortical computations over speech stimuli [9,10,25,27,29,42–45,79]. Our findings therefore suggest that low-frequency phase coherence contributes to the comprehension of syntactic information. Given that the acoustics of the phrase and sentence conditions were statistically indistinguishable (see Methods for acoustic normalization and analysis), the phase coherence difference may instead reflect the formation or processing of syntactic structures via endogenous recruitment of neural ensembles.

## Low-frequency (approximately <2 Hz) phase connectivity degree separates phrases and sentences

We initially calculated phase connectivity over the sensor space by ISPC at each time–frequency bin (for details, see Methods). We then used a statistical threshold to transform each connectivity representation to a super-threshold count at each bin. After baseline correction, we conducted a 1000-time cluster-based permutation test on a 3500-ms time window starting at the audio onset and over the frequencies from 1 Hz to 8 Hz to compare the degree of the phase connectivity between phrases and sentences (for details, see Methods). Phrases and sentences showed a significant difference in connectivity ($p < 0.01$ **, 2-tailed). The effect corresponded to a cluster extended from approximately 1800 ms to approximately 2600 ms after the speech stimulus onset and was mainly located at a very low frequency range (approximately <2 Hz). In the selected latency and frequency range, the effect was most pronounced at the right posterior region.

Fig 3A shows the temporal evolution of the separation effect, which is represented by the connectivity degree of the phrase condition minus the connectivity degree of sentence condition (in steps of 100 ms). Fig 3B shows the time–frequency decomposition of the phase connectivity degree, which is averaged across all sensors in the cluster. The left and right panel are the time–frequency plot for the phrase condition and the sentence condition, respectively.

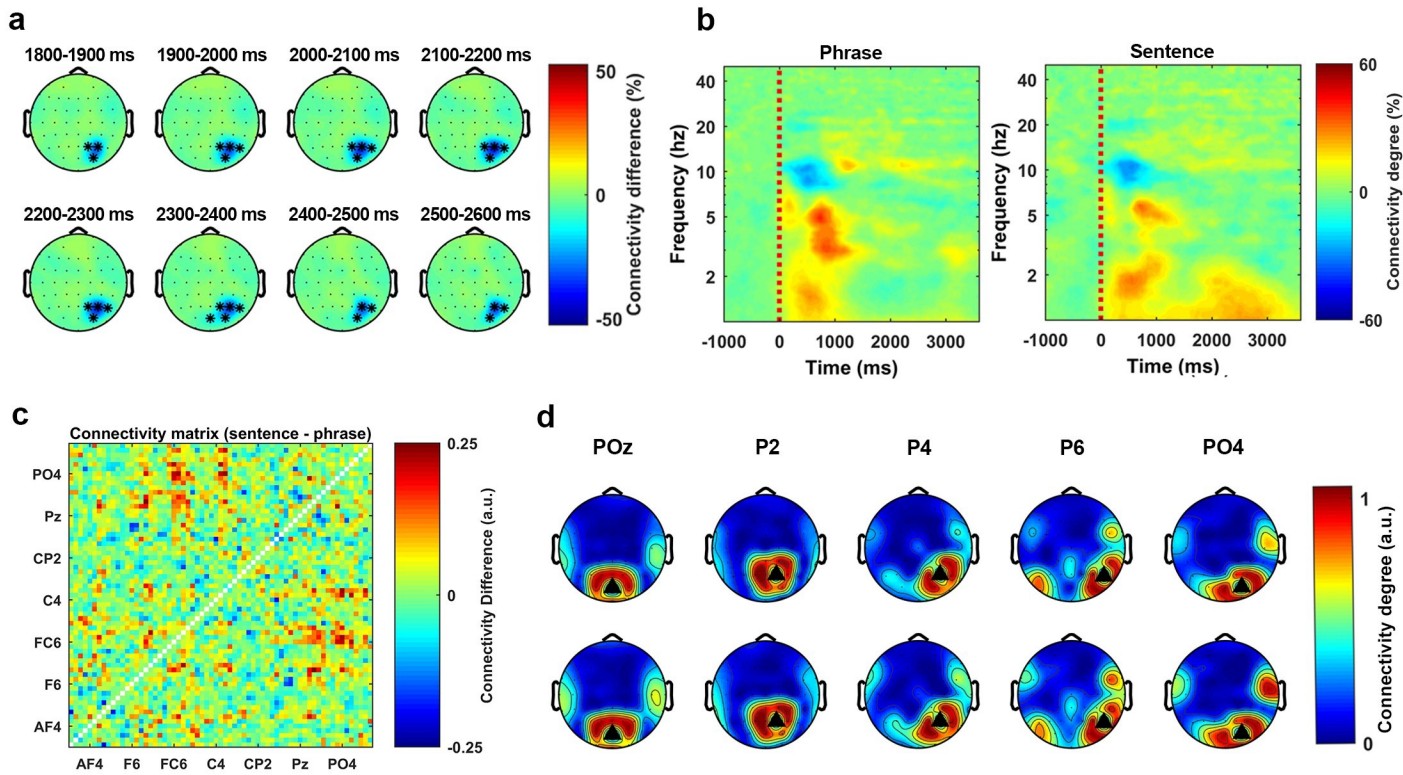

**Fig 3. Low-frequency phase connectivity separates phrases and sentences.** Statistical analysis on the phase connectivity degree was conducted using a nonparametric cluster-based permutation test (1000 times) on a 3500-ms time window, which started at the audio onset and over the frequencies from 1 Hz to 8 Hz. The results indicated that the phase connectivity degree was higher for the sentences than the phrases ($p < 0.01$**, 2-tailed). **(a)** The temporal evolution of the cluster. The activity was drawn by using the averaged connectivity degree of the phrases minus the connectivity degree of the sentences. The topographies were plotted in steps of 100 ms. **(b)** The time–frequency decomposition of the connectivity degree, which was averaged over all the sensors in the cluster. The left and the right panel show the connectivity degree of the phrase condition and the sentence condition, respectively. **(c)** The matrix representation of the phase connectivity differences between the phrases and the sentences. The figure was drawn by using the averaged connectivity matrix of the sentences minus the averaged connectivity matrix of the phrases. **(d)** All the sensors in this cluster were used as the seed sensors to plot the topographical representation of the phase connectivity. The upper panel and the lower panel show the phase connectivity of the phrases and the sentences, respectively.

Since the statistical analysis indicated a difference in degree of phase connectivity between phrases and sentences, we assessed how this effect was distributed in the sensor space. To do so, we extracted all binarized connectivity matrices that corresponded to the time and frequency range of the cluster and averaged all the matrices in this range for both conditions (for details, see Methods). **Fig 3C** shows the averaged matrix representation of the sentence condition minus the averaged matrix representation of the phrase condition. This result suggests that the connectivity difference was mainly localized at the frontal–central area. After extracting the matrix representation, we used all sensors of this cluster as seeds to plot connectivity topographies for both conditions. **Fig 3D** shows the pattern of the thresholded phase connectivity. The black triangles represent the seed sensors. The upper panel and lower panel represent the phrase and the sentence condition, respectively. The figure shows how the phase connectivity (synchronization) is distributed on the scalp in each condition. From this figure we can see that the overall degree of the phase connectivity was stronger for the sentence condition than the phrase condition.

The analysis indicated that the phase connectivity degree over the sensor space at the low-frequency range (approximately <2 Hz) could reliably separate the two syntactically different stimuli and that the effect was most prominent at the right posterior region.

## PAC as a generalized neural mechanism for speech perception

To assess whether phase amplitude coupling (PAC) distinguished phrases and sentences, we calculated the PAC value at each phase–amplitude bin for each condition and then transformed it to the PAC-Z (for details, see Methods). The grand average (average over sensors, conditions and participants) of the PAC-Z showed a strong activation over a region from 4 Hz to 10 Hz for the frequency of phase and from 15 Hz to 40 Hz for the frequency of amplitude. We therefore used the averaged PAC-Z value in this region as the region of interest (ROI) for sensors clustering. For each participant, we first selected 8 sensors that had the highest PAC-Z (conditions averaged) at each hemisphere. Averaging over sensors was conducted separately for the conditions (phrase and sentence) and the two hemispheres (see **Fig 4A**). The Bonferroni correction was performed to address the multiple comparison problem. This resulted in the z-score of 3.73 for $p$-value equals 0.05 (the z-score corresponded to the $p$-value equals 0.05 divided 11 (the number of phase bins) *12(the number of amplitude bins) *4(the number of conditions)). From the results, we can see that there was a strong low-frequency phase (4 Hz to 10 Hz) response entrained to high frequency amplitude (15 Hz to 40 Hz). The results indicate that the PAC was introduced when participants listened to the speech stimuli.

**Fig 4B** shows how the sensors were selected. The larger the red circle indicates the more often the sensor was selected across participants.

**Fig 4C** shows the topographical representation of the PAC-Z. The activity in these figures was the averaged PAC-Z values over the ROI. The results indicate that when the participants listened to the speech stimuli, PAC was introduced symmetrically at both hemispheres over the central area. This could be evidence for the existence of PAC when speech stimuli are being processed. However, both the parametric and the nonparametric statistical analysis failed to show a significant difference of the PAC-Z between phrases and sentences, which means we do not have evidence to show that the PAC was related to syntactic information processing. Therefore, our results suggest the PAC could be a generalized neural mechanism for speech perception, rather than a mechanism specifically recruited during the processing of higher-level linguistic structures.

## Alpha-band inhibition reflects discrimination between phrases and sentences

To query whether neural oscillations at the alpha band reflect the processing of syntactic structure, we calculated the induced power. The grand average (over all participants and all conditions) of the

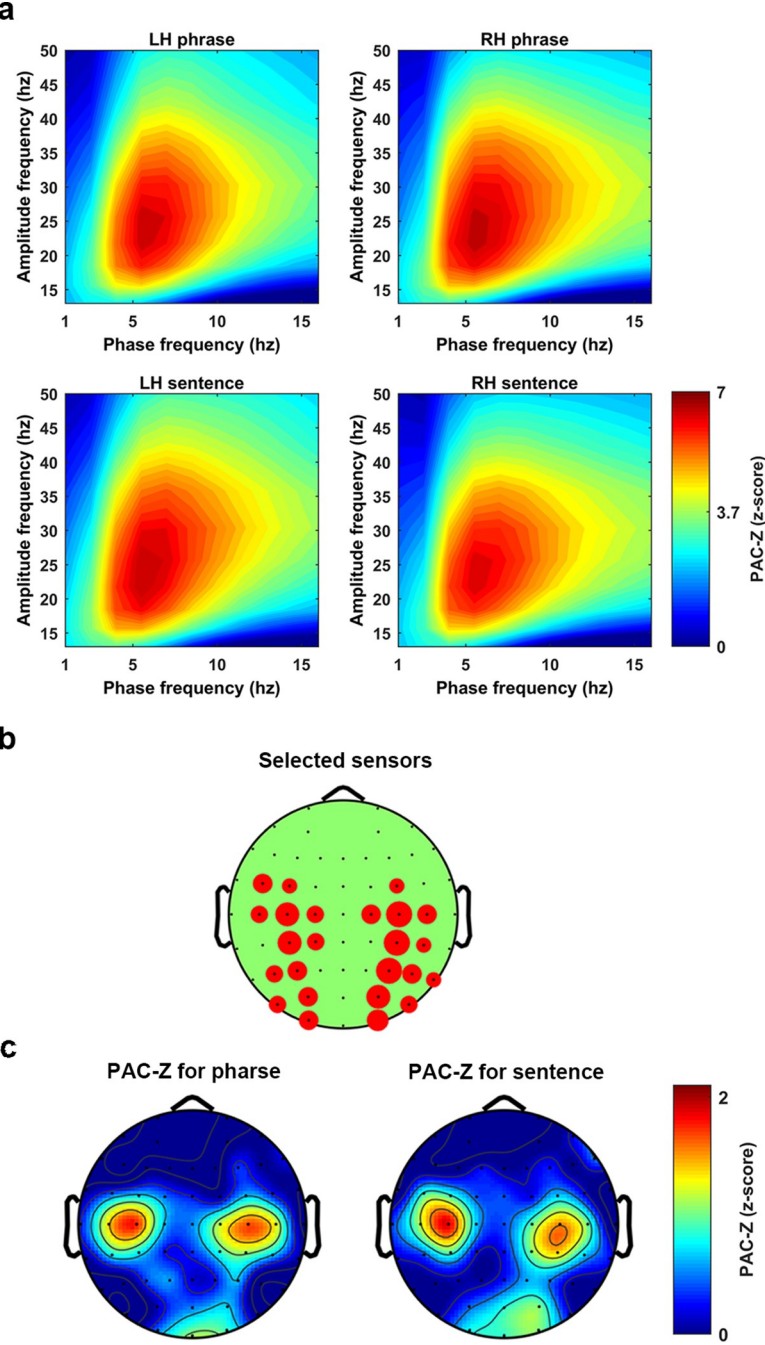

**Fig 4. PAC as a general mechanism for speech perception.** The figure shows a z-score transformed PAC, PAC-Z. **(a)** The PAC-Z for the phrases and the sentences at each hemisphere. Each figure was created by averaging 8 sensors which showed the biggest PAC-Z over the ROI. A z-score transformation with Bonferroni correction was conducted to test the significance, which lead to the threshold to be 3.73 corresponding to *p*-value equals 0.05. **(b)** The figure shows how sensors were selected at each hemisphere. The bigger the red circle indicates the more times this sensor was selected across participants. **(c)** The topographical distribution of the PAC-Z, which indicates the PAC was largely localized at the bilateral central areas. PAC, phase amplitude coupling; ROI, region of interest.

induced power showed a strong inhibition at the alpha band (approximately 7.5 to 13.5 Hz). Therefore, we checked whether this alpha-band inhibition could separate the two types of linguistic structures. Statistical analysis was conducted using a nonparametric cluster-based permutation test (1000

times) over the frequencies of alpha band with a 1000 ms time window that started at the audio onset (for details, see Methods). The results indicated that the alpha-band inhibition was stronger for the phrase condition than the sentence condition ($p < 0.01$ **, 2-tailed). In the selected time and frequency range, the effect corresponded to a cluster that lasted from approximately 350 ms to approximately 1,000 ms after the audio onset and was largely localized at the left hemisphere, although the right frontal–central sensors were also involved during the temporal evolution of this cluster. **Fig 5A** shows the temporal evolution of this cluster in steps of 50 ms using the induced power of the phrase condition minus the induced power of the sentence condition. **Fig 5B** shows the time–frequency plot of the induced power using the average of all the sensors in this cluster. The upper and lower panel shows the phrase condition and the sentences condition, respectively. From these figures, we can see that the alpha-band inhibition was stronger for the phrase condition than the sentence condition. These results show that the processing of phrases and sentences is reflected in the intensity of the induced neural response in the alpha band.

## The power connectivity in the alpha band indicates a network level separation between phrases and sentences

We calculated power connectivity in each sensor-pair at each time–frequency bin using Rank correlation (for details, see Methods). The grand average (over all participants and all

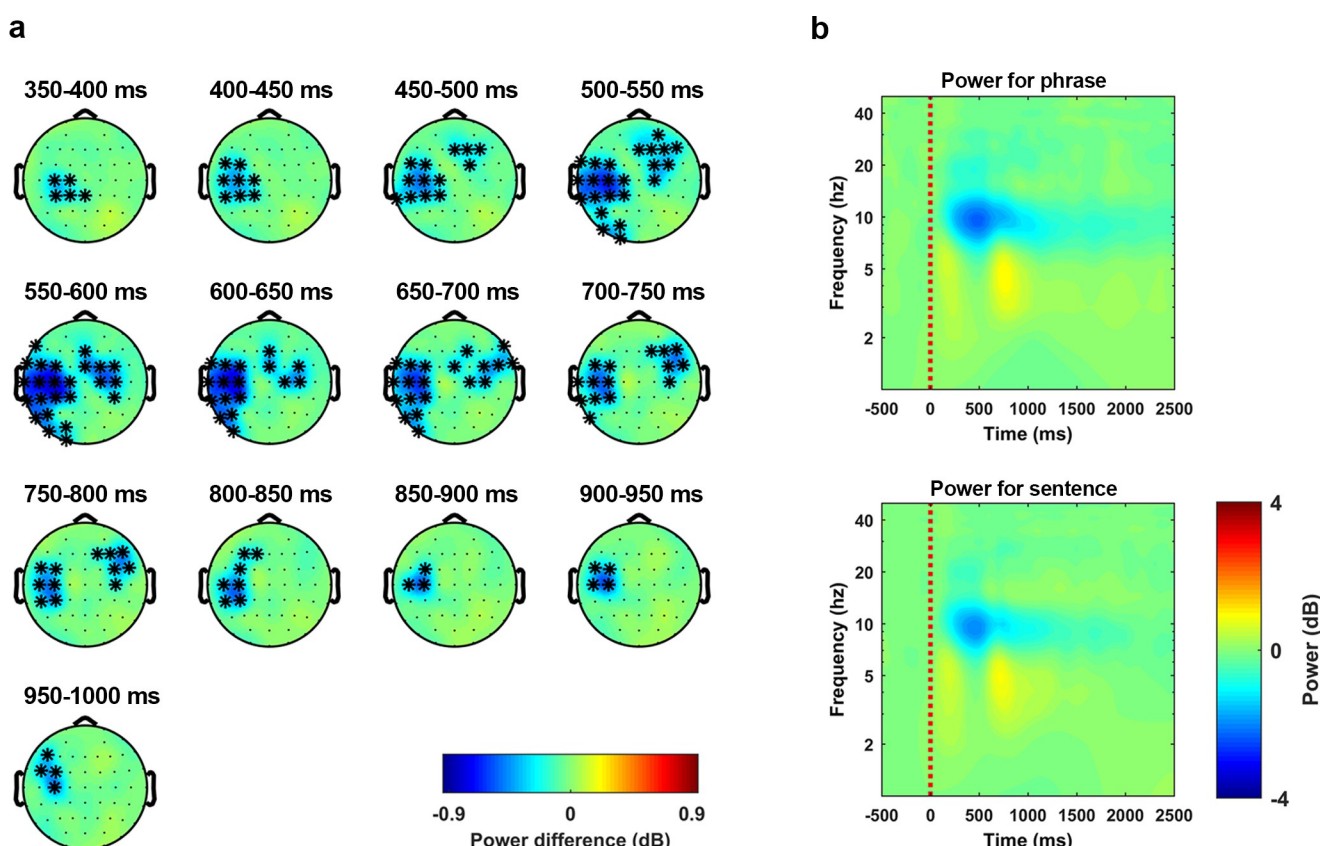

**Fig 5. Alpha-band inhibition suggests a separation between phrases and sentences.** Statistical analysis on the induced activity was conducted using a nonparametric cluster-based permutation test (1000 times) on a 1000-ms time window, which started at the audio onset and over the frequencies from 7.5 Hz to 13.5 Hz. The results indicated that the power was higher for sentences than phrases ($p < 0.01$ **, 2-tailed). **(a)** The temporal evolution of the cluster that corresponds to the discrimination effect. The activity was drawn by using the averaged induced power of the phrases minus the averaged induced power of the sentences. The topographies were extracted in steps of 50 ms. **(b)** Induced power averaged over all the sensors in this cluster. The upper panel and the lower panel show the induced power of the phrases and the sentences, respectively.

conditions) of the power connectivity level showed a strong inhibition at the alpha band from 100 ms to 2200 ms after the audio onset. This region, which showed a strong power connectivity inhibition, was defined as the ROI. For each participant, we selected 8 sensors at each hemisphere that showed the biggest inhibition on the condition averaged power connectivity. Averaging across all the selected sensors was followed up, which resulted in 4 conditions for each participant (left phrase, left sentence, right phrase, and right sentence).

Fig 6A shows the power connectivity degree which was averaged over all participants for each condition. To check whether power connectivity degree could separate the phrases and the sentences, a Stimulus-Type*Hemisphere two-way repeated measures ANOVA was conducted. The comparison revealed a main effect of Stimulus-Type (F (1, 14) = 5.28, $p$ = 0.033 *). Planned post-hoc comparisons using paired sample $t$-tests on the main effect of Stimulus-Type showed that the power connectivity inhibition was stronger for the phrases than the sentences (t (29) = 2.82, $p$ = 0.0085 **). Fig 6B shows the power connectivity degree for each extracted condition. Fig 6C shows how sensors were selected. The size of the red circle indicates the more times a sensor was selected.

Since the degree of the power connectivity over the alpha-band indicated a separation between phrases and sentences, we also checked how this difference was distributed in the sensor space. To do so, we extracted the binarized power connectivity representations (matrices) that are located in the ROI, then averaging was performed for each condition across all

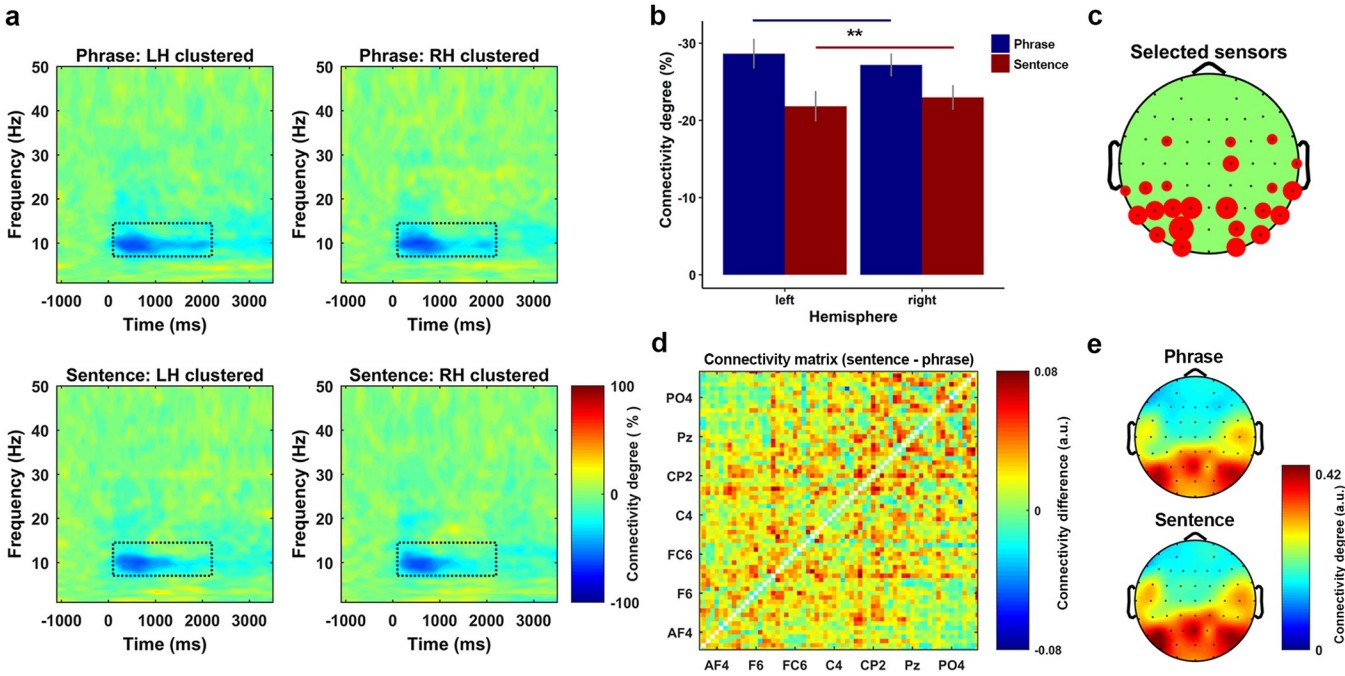

**Fig 6. Power connectivity in the alpha band suggests a separation between phrases and sentences. (a)** Power connectivity degree for all conditions. Each plot was clustered by the sensors at each hemisphere that showed the biggest inhibition on the grand averaged power connectivity. **(b)** The results of a two-way repeated measures ANOVA for the power connectivity on the factors of stimulus-type (phrase or sentence) and hemisphere (left or right). The results indicate a significant main effect of stimulus-type, post-hoc comparison on the main effect indicated that the overall inhibition level of the power connectivity was stronger for the phrases than the sentences (t (29) = 2.82, $p$ = 0.0085 **, two-sided). **(c)** How sensors were selected for the clustering. The size of the red circle indicates the more times the sensor was selected across participants. **(d)** The connectivity differences between the phrases and the sentences on all sensor pair. The figure was drawn using the average of the binarized connectivity matrix of the sentences minus the matrix of the phrases. The results indicate that the connectivity degree over the sensor space for the sentences was higher than the phrases. **(e)** Topographical representation of the binarized connectivity, which was clustered using the sensors showed biggest inhibition on the power connectivity. The upper and lower panel shows the phrase and sentence condition, respectively.

connectivity matrices. **Fig 6D** shows the difference of the power connectivity degree over the sensor space using the average of the binarized sentence connectivity matrix minus the average of the binarized phrase connectivity matrix. The results indicate that the inhibition of power connectivity was stronger for phrases than for sentences. In other words, the overall level of power connectivity was higher for sentences than for phrases. **Fig 6E** is the topographical representation of the power connectivity, which was plotted using the binarized power connectivity of the selected sensors. The upper and lower panel are the phrase condition and the sentence condition, respectively. From this figure, we can see that the difference was largely localized at the bilateral central area and more strongly present at the left than the right hemisphere. These results reflect that the neural network which was organized by the intensity of the induced power at the alpha band was different for phrases and sentences.

## Different encoding states for phrases versus sentences in both temporal and spectral dimensions

Previous research has shown that the low-frequency neural response reliably reflects the phase-locked encoding of the acoustic features of speech [72,73]. Therefore, we initially tested whether the neural response from all canonical frequency bands could equally reflect the encoding of the acoustic features. To do so, we fitted the STRF for each condition at all frequency bands, which are Delta (<4 Hz), Theta (4 to 7 Hz), Alpha (8 to 13 Hz), Beta (14 to 30 Hz), and Low Gamma (31 to 50 Hz), respectively. Then we compared the real performance of the STRFs to the random performance of them (for details, see Methods). **Fig 7A** shows the results of this comparison. The blue and red dots represent the real performance of the STRFs, the error bar presents 1 SEM on each side. The small gray dots represent the random performance (1000 times in each frequency band per condition). The upper boarder, which is delineated by these gray dots represents the 97.5 percentiles of the random performance. The performance of the STRFs was above chance level only at low frequencies (delta and theta), which is consistent with previous research [72,73]. Our results confirm that the low-frequency STRF reliably reflects the relationship between the acoustic features of speech and the neural response at low frequencies.

Since only low-frequency neural responses robustly reflected the encoding of the speech stimuli, we fitted the STRF for both conditions using the neural response that was low-pass filtered at 9 Hz. Leave-one-out cross-validation was used to maximize the performance of the STRFs (for details, see Methods). **Fig 7B** shows the performance of the STRF for each condition. The transparent dots, blue for phrases and red for sentences, represent the model's performance on each testing trial. The solid dots represent the model's performance that was averaged over all trials, the error bars represent one standard error of the mean (SEM) on each side of the mean. A paired sample $t$-test was used to compare the performance between the phrase and sentence conditions. No evidence was found to indicate a performance difference between the two conditions (t (74) = 1.25, $p$ = 0.21). The results indicated that the STRF were fitted equally well for phrases and sentences. Thus, any difference in temporal–spectral features between the STRF of phrases and sentences cannot be driven by the model's performance. **Fig 7C** shows the comparison between the real neural response and the model predicted response at the sample sensor Cz. The upper and lower panel shows the performance of the STRF to phrases (r = 0.47, $N$ = 1,024, $p$ < 1e-5 ***) and sentences (r = 0.41, $N$ = 1,024, $p$ < 1e-5 ***), respectively.

The grand average of the STRFs was negative from 0 to 400 ms in the time dimension and from 100 to 1000 Hz in the frequency dimension, and the sensor clustering of the STRF was conducted based on the averaged activation on this ROI. More concretely, we selected 8 sensors at each hemisphere for each participant, which showed the strongest averaged magnitude

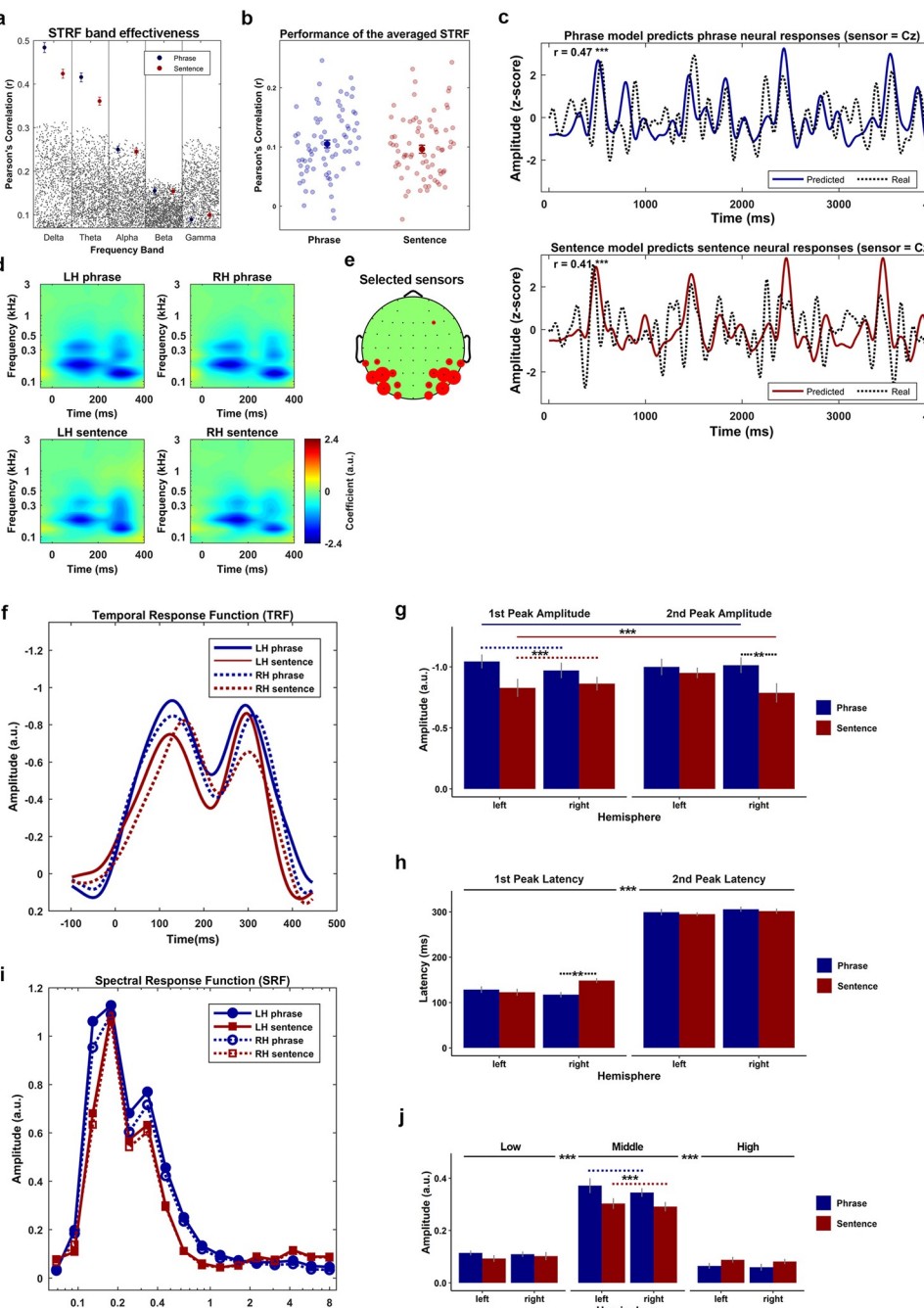

**Fig 7. Acoustic features are encoded differently between phrases and sentences in a phase locked manner. (a)** Comparison between the real performance and the random performance of the STRF in each canonical frequency band. The results suggested that only the performance of the STRF in the delta band (<4 Hz) and theta band (4 to 8 Hz) was statistically better than the random performance. The blue and red dots represent the real performance of the STRFs for the phrases and the sentences, respectively. The error bar represents two SEM centered on the mean. The gray dots represent the random performance drawn by permutations. **(b)** The performance of the low-frequency range (<8 Hz) STRF averaged across all participants. The solid blue and red dot represent the averaged performance across all testing trials. The error bar represents two SEM across the mean. The transparent blue and red dots represent the model's performance on each testing trial for the phrases and the sentences, respectively. The results indicated no performance difference on the kernel between the phrases and the sentences. **(c)** The comparison between the real neural response (dashed lines) and the model predicted response (solid blue for the phrase, solid red for the sentence) at a sample sensor Cz. The results suggest that the STRFs performed equally well for the phrases (r = 0.47, *p* < 1e-5 ***, *n* = 1,024) and the sentences (r = 0.41, *p* < 1e-5 ***, *n* = 1,024). **(d)** The clustered STRF using the selected sensors

showing the biggest activity (negative) on the ROI. The figure on the left and right side of the upper panel represents the clustered STRF for the phrases at the left and right hemisphere, respectively. The corresponding position of the lower panel represents the clustered kernel for the sentences. **(e)** The figure shows how the sensors were selected, in which the bigger the red circle represents the more times the sensor was selected across all participants. **(f)** The TRFs that were decomposed from the STRFs, in which the blue and red lines represent the phrases and the sentences, respectively. The solid and the dashed lines represent left and right hemisphere, respectively. **(g)** The comparison of the magnitude of the TRFs. The blue and the red bars represent the phrases and the sentences, respectively. The error bar shows 1 SEM across the mean on each side. A 3-way repeated measure ANOVA of the peak magnitude was conducted on the factors of Stimulus-type (phrase or sentence), Hemisphere (left or right), and Peak-type (approximately 100 ms or approximately 300 ms). The results indicated a main effect of Stimulus-type and a 3-way interaction. The post hoc comparison on the main effect of Stimulus-type suggested that the amplitude (negative) was stronger for the phrase condition than the sentence condition (t (59) = 4.55, $p <$ 2e-5 ***). To investigate the 3-way, Stimulus-type*Peak-type*Hemisphere, interaction, two 2-way repeated measure ANOVA with the Bonferroni correction were conducted on the factors of Hemisphere and Stimulus-type at each level of the Peak-type. The results indicated a main effect of Stimulus-type at the first peak (F (1, 14) = 8.19, $p$ = 0.012 *) and a 2-way Hemisphere*Stimulus-type interaction at the second peak (F (1, 14) = 6.42, $p$ = 0.023 *). At the first peak, a post hoc comparison on the main effect of Stimulus-type was conducted using a paired sample $t$ tests, the results showed that the magnitude of the phrase condition was higher than the magnitude of the sentence condition (t (29) = 3.49, $p$ = 0.001 ***). For the 2-way, Hemisphere*Stimulus-type, interaction at the second peak, the paired sample t tests with Bonferroni correction was conducted to compare the difference of the magnitude between the phrases and the sentences at each hemisphere. The results indicate that the magnitude at the second peak was stronger for the phrase condition than the sentence condition in the right hemisphere (t (14) = 3.21, $p$ = 0.006 **), but not the left hemisphere (t (14) = 0.86, $p$ = 0.40). **(h)** The comparison of the peak latency of TRFs, the blue and the red bars represent the phrases and the sentences, respectively. The error bar shows 1 SEM across the mean on each side. A 3-way repeated measure ANOVA of the peak latency was conducted on the factors of Stimulus-type (phrase or sentence), Hemisphere (left or right) and Peak-type (approximately 100 ms or approximately 300 ms). The results indicated a main effect of Peak-type and a 3-way interaction. The post hoc comparison on the main effect of Peak-type suggested that the latency of the first peak was significantly faster than the second peak (t (59) = 38.89, $p <$ 2e-16 ***). The post hoc comparison on the 3-way interaction with the Bonferroni correction on the factors of Hemisphere and Stimulus-type for each level of the Peak-type suggested a 2-way Hemisphere*Stimulus-type interaction at the first peak (F (1, 14) = 12.83, $p$ = 0.002**). The post hoc comparison on this 2-way interaction using paired sample t tests with the Bonferroni correction indicated that the latency at the first peak was significantly longer for the sentences than the phrases at the right hemisphere (t (14) = 3.55, $p$ = 0.003 **), but not the left hemisphere (t (14) = 0.58, $p$ = 0.56). **(i)** The SRFs which were decomposed from the STRFs, in which the blue and the red lines represent the phrases and the sentences, respectively, the solid and the dashed lines represent the left and right hemisphere, respectively. **(j)** The comparison of the amplitude of the SRFs. The SRF was first separated into 3 bands, low ($<$0.1 kHz), middle (0.1 to 0.8 kHz) and high ($>$0.8 kHz) based on the averaged frequency response of the STRF, then a 3-way repeated measure ANOVA of the amplitude was conducted on the factors of Stimulus-type (phrase or sentence), Hemisphere (left or right) and Band-type (low, middle, and high). The results indicated a main effect of Band-type (F (2, 28) = 119.67, $p <$ 2e-14 ***) and a 2-way, Band-type*Stimulus-type, interaction (F (2, 28) = 27.61, $p <$ 3e-7 ***). The post hoc comparison on the main effect of Band-type using paired sample t tests with Bonferroni correction showed that the magnitude of the middle frequency band was stronger than the low-frequency band (t (59) = 17.9, $p <$ 4e-25 ***) and high frequency band (t (59) = 18.7, $p <$ 5e-26 ***). The post hoc comparison using paired sample t tests with the Bonferroni correction on the, Band-type*Stimulus-type, interaction showed that the amplitude of the SRF was stronger for the phrases than the sentences only at middle frequency band (t (29) = 4.67, $p <$ 6e-5 ***). ROI, region of interest; STRF, spectral temporal response function; TRF, temporal response function.

(negative) in this region. **Fig 7D** shows the clustered STRFs that were averaged across all participants. **Fig 7E** shows how the sensors were selected across the participants, in which the bigger the red circle indicates the more times a given sensor was selected.

To compare the differences of the kernel (STRF) in both the temporal and spectral dimensions, the temporal response function (TRF) and the spectral response function (SRF) were extracted separately for each condition (for details, see Methods).

**Fig 7F** shows the TRFs that were averaged across all participants. The grand average of all TRFs showed 2 peaks at approximately 100 ms and approximately 300 ms. We therefore defined the first temporal window from 50 to 150 ms (center at 100 ms) and the second temporal window from 250 to 350 (center at 300 ms) for searching the magnitude and the latency of these 2 peaks. The latency of each peak was defined as the time when it appeared. The magnitude of each peak was defined as the average magnitude over a 5 ms window on both sides around it. After extracting the magnitude and the latency of these 2 peaks, a Stimulus-

type*Peak-type*Hemisphere 3-way repeated measures ANOVA was conducted on both the magnitude and the latency.

For the magnitude of the TRF (**Fig 7G**), the statistical comparison showed a significant main effect of Stimulus-type (F (1, 14) = 13.58, $p$ = 0.002 **) and a significant three-way, Stimulus-type*Peak-type*Hemisphere, interaction (F (1, 14) = 15.25, $p$ = 0.001 ***).

The post-hoc comparison on the main effect of Stimulus-Type using paired-sample $t$ tests showed that the magnitude for phrases was significantly larger than the magnitude for sentences (t (59) = 4.55, $p$ < 2e-5 ***). This result suggests that the instantaneous neural activity in response to phrases had a stronger phase-locked dependency on the acoustic features than in response to sentences.

To investigate the three-way, Stimulus-type*Peak-type*Hemisphere, interaction, two two-way repeated measures ANOVAs with the Bonferroni correction were conducted on the factors of Hemisphere and Stimulus-type at each level of the Peak-type. The results indicated a main effect of Stimulus-Type at the first peak (F (1, 14) = 8.19, $p$ = 0.012 *) and a two-way Hemisphere*Stimulus-Type interaction at the second peak (F (1, 14) = 6.42, $p$ = 0.023 *).

At the first peak, we conducted a post-hoc comparison on the main effect of Stimulus-type using a paired sample $t$-tests, which showed that the magnitude of the phrase condition was higher than the magnitude of the sentence condition (t (29) = 3.49, $p$ = 0.001 ***). The results indicate that the instantaneous neural activity was more strongly driven by the acoustic features that were presented approximately 100 ms ago when phrases were presented compared to when sentences were presented.

For the two-way, Hemisphere*Stimulus-Type, interaction at the second peak, the paired- sample $t$-tests with Bonferroni correction was conducted to compare the difference of the magnitude between phrases and sentences at each hemisphere. The results indicated that the magnitude at the second peak was stronger for phrases than sentences in the right hemisphere (t (14) = 3.21, $p$ = 0.006 **), but not the left hemisphere (t (14) = 0.86, $p$ = 0.40). The findings suggest that, at the right hemisphere, the instantaneous neural activity of the phrases was more strongly driven by the acoustic features that were present approximately 300 ms than it was under sentences.

For the latency of the TRF (**Fig 7H**), the comparison showed a main effect of the Peak-type (F (1, 14) = 1e+3, $p$ < 1e-14 ***) and a three-way, Stimulus-type*Peak-type*Hemisphere, interaction (F (1, 14) = 8.04, $p$ = 0.013 *).

The post-hoc comparison for the main effect of the Peak-type with paired-sample t-tests showed, as expected, that the latency of the first peak was significantly shorter than the second one (t (59) = 38.89, $p$ < 2e-16 ***). The result is clear since regardless of search method for the analysis time windows, the latency of the first peak will always be earlier than the second peak.

To investigate the 3-way, Stimulus-type*Peak-type*Hemisphere, interaction, two two-way repeated measures ANOVA with the Bonferroni correction were conducted on the factors of Hemisphere and Stimulus-type for each level of the Peak-type. The comparison suggested a two-way Hemisphere*Stimulus-Type interaction at the first peak (F (1, 14) = 12.83, $p$ = 0.002 **). The post hoc comparison on this two-way interaction using paired-sample $t$-tests with the Bonferroni correction indicated that the latency at the first peak was significantly later for sentences than for phrases at the right hemisphere (t (14) = 3.55, $p$ = 0.003 **), but not the left hemisphere (t (14) = 0.58, $p$ = 0.56). The results suggest that, within the first temporal window (approximately 50 to 150 ms), only at the right hemisphere, the neural response to sentences was predominantly driven by acoustic features that occurred earlier in time than the acoustic features that drove the neural response of the phrases.

**Fig 7I** shows the SRFs that were averaged across all participants. The grand average of the STRFs suggested that the activation of the kernel was most prominent in the frequency range from 0.1 kHz to 0.8 kHz. To compare the differences of the neural encoding of the acoustic

features in the spectral dimension, we separated the SRF into 3 frequency bands, which were lower than 0.1 kHz, 0.1 to 0.8 kHz and higher than 0.8 kHz. We then averaged the response in each extracted frequency band for each condition. The statistical comparison was conducted using three-way repeated measures ANOVA on the factors of Hemisphere, Stimulus-type and Band-type. The results (**Fig 7J**) indicated a main effect of Band-type (F (2, 28) = 119.67, $p < $ 2e-14 ***) and a 2-way, Band-type*Stimulus-type, interaction (F (2, 28) = 27.61, $p < $ 3e-7 ***).

The post-hoc comparison on the main effect of Band-type using paired-sample *t*-tests with Bonferroni correction showed that the magnitude of the middle frequency band was stronger than that of the low-frequency band (t (59) = 17.9, $p < $ 4e-25 ***) and high-frequency band (t (59) = 18.7, $p < $ 5e-26 ***). The results indicated that the acoustic features from different frequency bands contributed differently to the evoked neural response. In other words, for both conditions, the neural response was predominantly driven by the encoding of acoustic features from 0.1 kHz to 0.8 kHz, which was considered as the spectral–temporal features at the range of the first formant [80–83].

The post hoc comparison using paired-sample *t*-tests with the Bonferroni correction on the, Band-type*Stimulus-Type, interaction showed that the amplitude of the SRF was stronger for the phrase condition than the sentence condition only at the middle frequency band (t (29) = 4.67, $p < $ 6e-5 ***). The results indicate that at the middle frequency range, the neural response of phrases was more strongly predicted solely by modeling the encoding of the acoustic features than it was in the sentence condition. This pattern of results indicates that the neural representation of sentences is more abstracted away from the neural response that is driven by the physicality of the stimulus.

## Methods

### Participants

A total of 15 right-handed Dutch native speakers, 22 to 35 years old, 7 males, participated in the study. All participants were undergraduate or graduate students. Participants reported no history of hearing impairment or neurological disorder and were paid for their participation. The experimental procedure was approved by the Ethics Committee of the Social Sciences Department of Radboud University. Written informed consent was obtained from each participant before the experiment.

### Stimuli

We selected 50 line drawings of common objects from a standardized corpus [84]. The Dutch names of all objects were monosyllabic and had nonneuter or common lexical gender. In our experiment, the objects appeared as colored line drawing on a gray background. More specifically, we presented each line drawing in 5 colors: blue (blauw), red (rood) yellow (geel), green (groen), and purple (paars). In total, this yielded 250 pictures. The line drawings were sized to fit into a virtual frame of 4 cm by 4 cm, corresponding to 2.29˚ of visual angle for the participant.

We then selected 10 figures with different objects in each color, without object replacement between colors, to create speech stimuli. For each selected line drawing, a 4-syllable, phrase–sentence pair was created, e.g., *De rode vaas (The red vase) and De vaas is rood (The vase is red)*. Note that sentences contain the verb *is* while phrases contain an inflectional agreement morpheme *-e* on the word *rood* in order to be grammatical in Dutch (*rode*); these different properties lead to morphemic and syntactic differences between conditions. In total, we had 100 speech stimuli (50 phrases and 50 sentences). All stimuli were synthesized by a Dutch male voice, "Guus", of an online synthesizer (www.readspeaker.com), and were 733 ms to

1,125 ms in duration (Mean = 839 ms, SD = 65 ms). See S1 **Appendix** for all the stimuli that were used in this experiment.

## Acoustic normalization and analysis

In order to normalize the synthesized auditory stimuli, they were first resampled to 44.1 kHz. Then all speech stimuli were adjusted by truncation or zero padding at both ends to 1000 ms without missing any meaningful dynamics. Then 10% at both ends of each stimulus was smoothed by a linear ramp (a cosine wave) for removing the abrupt sound burst. Finally, for controlling the intensity of speech stimuli, the root mean square (RMS) value of each stimulus was normalized to −16 dB.

The intensity fluctuation of each speech stimulus was characterized by the corresponding temporal envelope, which was extracted by the Hilbert transform of the half-wave rectified speech signal. Then each extracted temporal envelope was downsampled to 400 Hz. For checking the acoustic properties in frequency domain, the discrete Fourier transform (DFT) was performed to extract the spectrum of the temporal envelope. Decibel transformation for the spectrum of each speech stimulus was performed by using the highest frequency response in the corresponding phrase–sentence pair as the reference.

**Fig 8A** and **8B** shows the syntactic representation of the phrases and sentences. Since all the phrases, or all the sentences, have the same syntactic structure, we selected a sample pair, *De rode vaas (The red vase) and De vaas is rood (The vase is red)*, to show the syntactic decomposition. Four syllables were strictly controlled to be the physical input for both conditions. The syntactic structure relations within conditions, which describe how words are combined into a meaningful linguistic structure, as well as the morphemes that cue these relations, are different. The syntactic structure for the sentence condition contains more and different syntactic constituents, which, in turn, stand in different hierarchical relationships than in the phrase condition; these differences are likely cued by the verb *is* and the morpheme *-e*. **Fig 8C** and **8D** shows the spectrogram of a sample phrase–sentence pair. The comparison suggests a similar temporal–spectral pattern in this sampled pair. **Fig 8E** show the temporal envelopes of this sample pair, the blue line for the phrase and the red line for the sentence, respectively. The comparison suggests a highly similar energy fluctuation between the phrase and the sentence. **Fig 8f** shows the intensity relationship of this sample pair in each frequency bin. The Pearson correlation was calculated to reveal the similarity between the spectrum of this sample pair (r = 0.94, $p < 1e-4$ ***). The comparisons indicated that they are highly similar in acoustic features. In this figure, the darker the dots represent the lower the frequency of the spectrum. **Fig 8G** shows the averaged temporal envelope across all the stimuli ($N = 50$), the blue line and red line represents the phrases and the sentences, respectively. The shaded areas cover 2 SEM centered on the mean. To check the similarity of the instantaneous intensity on the temporal envelopes between the phrases and the sentences, we first calculated the cosine similarity. For each time bin (400 bins in total), the similarity measure simultaneously treats the activity of all stimuli into one vector while considers each stimulus as one dimension (50 dimensions in total). To add signal to noise ratio (SNR), the energy fluctuation was averaged using a 50-ms window centered on each bin. Statistical significance was evaluated via a permutation approach. Specifically, we generated a reference distribution with 1000 similarity values, each of which was selected as the largest value of the cosine similarities that were calculated using the raw phrase envelopes with the time shuffled sentence envelopes. Our simulations suggested a threshold of 0.884 corresponding to the *p*-value of 0.05, as shown on the right-side vertical axis. The statistical analysis indicated a high similarity on the temporal dimension of the energy profile between the phrases and the sentences. **Fig 8H** shows the comparison between the averaged

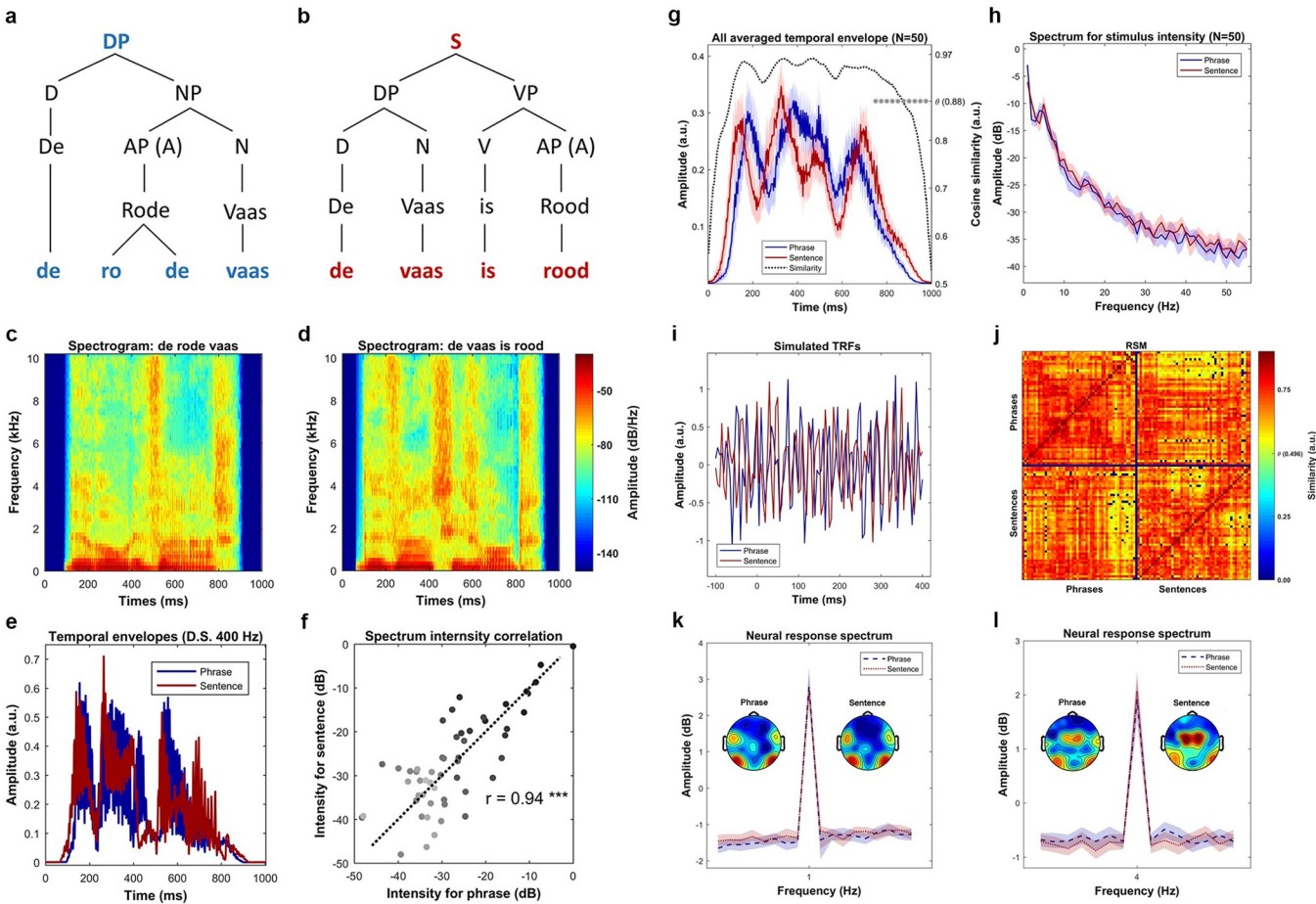

**Fig 8. Stimulus comparison between phrases and sentences. (a)** Syntactic structure of phrases, which is represented by a sample phrase, *de rode vaas (the red vase)*. The DP can first be decomposed into a determiner (D) and a NP, in which the NP can be separated into an AP, which equivalents to an adjective (A), and a noun (N). Finally, these words can be decomposed into 4 syllables. **(b)** Syntactic structure of sentences, which represented by a sample sentence, *de vaas is rood (the vase is red)*. The sentence can be decomposed into two parts, which are a DP and a VP, respectively. The DP can then be separated into a determiner (D) and a noun (N), and the VP can be separated into a combination between a verb (V) and an adjective (A). All these words are finally decomposed into four syllables. **(c)** and **(d)** shows the spectrogram of the sample phrase and the sample sentence, respectively. The comparison between the spectrograms indicates a similar pattern between these 2 types of stimulus. **(e)** shows the comparison of the temporal envelopes of the sample phrase–sentence pair, i.e., *De rode vaas (the red vase)* versus *De vaas is rood (the vase is red)*, which were down sampled to 400 Hz. The comparison suggests a similar energy profile of the sample pair. **(f)** shows the spectrum for the sample phrase–sentence pair, in which the horizontal axis and the vertical axis indicates the frequency response of the temporal envelop of the phrase and the sentence, respectively. The darker the dot indicates the higher the frequency. The Pearson correlation suggested that the spectrum is highly similar between the sample phrase and the sample sentence (r = 0.94, $p < 1e\text{-}5$ ***). **(g)** shows the averaged temporal envelope of these 2 types of stimuli, blue for phrase and red for sentence. The black dotted line indicates a highly similar physical properties between them in time domain by cosine similarity. Statistical analysis on the similarity measure using permutation test indicated an inseparable pattern. **(h)** Spectrum of the averaged envelopes for the two types of speech stimuli. The shaded area for each condition covers two SEM. across the mean (*N* = 50). Statistical analysis using Bayesian inference suggested a highly similar frequency response. **(i)** Shows the results of simulations using TRF. Statistical analysis using pairwise *t*-test indicated no difference between the two types of stimuli in every single time point, which suggests indistinguishable acoustics betwen the stimuli. **(j)** Similarity comparison for all possible stimuli pairs. As shown in the RSM, the upper-left and lower-right panel shows the comparison of all phrase-pair and all sentence-pair, respectively. The upper-right and lower-left matrix shows the comparison of all possible phrase–sentence pairs. Statistical comparison using permutation test indicated a highly similar acoustic properties across all possible pairs. **(k)** Shows the frequency tagging effect at 1 Hz. The figure shows a strong peak at 1 Hz for the phrases (t (14) = 8.72, $p < 4.9e\text{-}7$ ***) and the sentences (t (14) = 8.46, $p < 7.1e\text{-}7$ ***). It reflects that syntactic integration happened at 1 Hz, and our duration (1 second) normalization is effective. However, no difference of the 1-Hz activity was found between the conditions, which indicates a difficulty to separate the types of syntactic structures (t (14) = 0.63, $p = 0.53$) using frequency tagging approach. **(l)** shows the frequency tagging effect at 4 Hz. The strong 4 Hz peak for the phrases (t (14) = 7.79, $p < 1.8e\text{-}6$ ***) and the sentences (t (14) = 9.43, $p < 1.9e\text{-}7$ ***) suggests that syllables were the initial processing units for syntactic integration. AP, adjective phrase; DP, determiner phase; NP, noun phrase; TRF, temporal response function; VP, verb phrase.

spectrum of all phrases and the averaged spectrum of all sentences. The spectrum was considered to reflect the prosodic information of the speech stimulus [24,32,34,35,85]. In this figure, the shaded area covers two SEM across the stimuli. Statistical comparison using paired sample *t*-test was conducted at each frequency bin, in which no evidence was found to indicate significant physical difference between phrases and sentences. In addition, to show a statistically similar frequency response on the energy profiles between the phrases and the sentences, robust Bayesian inference on all frequency bins that above 1-Hz was conducted. Specifically, for each frequency bin, we first combined the instantaneous intensities across conditions into one pool. Then, a prior gamma-distribution for the mean of each condition, with the mean equals the average value of the pool and with the standard deviation equals five times the standard deviation of the pool, was generated. The normality for both conditions was governed by a constant value of 30. Posterior distribution was recurrently updated using Markov Chain Monte Carlo (MCMC), and the statistical significance was decided by whether zero is located in the 95% of highest density interval (HDI) of the posterior distribution for the difference of means. Robust Bayesian estimation allows to accept the null hypothesis when the 95% HDI are entirely located within the empirical range (−0.1 to 0.1) for the region of practical equivalence (ROPE) [86–93]. Our analysis on all frequency bins suggested that there is no difference on the spectral dimension of the envelopes (95% HDI located in the ROPE range from −0.1 to 0.1) between the phrases and the sentences. **Fig 8I** shows the simulation results using TRF. The reason for doing this is to show that any effect observed in this study is not driven by acoustic differences, and that the acoustic features are statistically-matched in the temporal dimension. The underlying assumption is that if the physical-acoustical properties of the phrases and sentences are similar, then fitting a kernel (TRF) using these speech stimuli with the same signal would give similar results. By testing this hypothesis, we fitted two TRFs for each condition 15 times (to imitate the number of participants), each time with 100 simulated acoustic-response pairs. The acoustic input was constructed by randomly selecting 15 speech stimuli in the corresponding condition; the simulated response was sampled from the standard Gaussian distribution. Optimization was performed using ridge regression and leave-one-out cross-validation (for details, see STRF section). After fitting the kernels, a paired-sample *t*-test on each time point was conducted, the comparison suggested no difference between the TRFs. Therefore, the simulation results also indicate statistically indistinguishable acoustic properties occurred across time between phrases and sentences.

As one might interested in the acoustic comparison not only on the targeted pairs (e.g., the pair with the same semantic components), but the pairs within conditions (e.g., a comparison between two phrases), we performed a similarity analysis on all the possible pairs in our stimuli. To do so, we first calculated the cosine similarity on the energy profile between any two of our stimuli, then depicted the results in a representational similarity matrix (RSM). Our analysis suggested a high similarity pattern, in which the mean similarity value was 0.74 (maximum 1, ranging from 0.43 to 0.94 when omit pairs with the same stimuli which equals 1). To test the statistical significance, we performed a 1000-time permutation test to form a null distribution. In each iteration, we calculated the cosine similarity between a randomly selected real envelope with the other randomly selected envelope that was shuffled in time. Our manipulations suggested a threshold of 0.496 corresponding to *p*-value of 0.05. The results indicated that 98.91% of all pairs were statistically similar. Note that only one targeted pair (i.e., a pair with controlled semantic components) did not reach the threshold. The results are shown in **Fig 8J**, in which the pairs with the similarity values lower than the threshold are labeled in dark blue squares. Note that the dark blue cross that separates the RSM into four regions were served as reference lines, which was only used for checking purposes (no data points are located in there).

In order to check whether syllables were the initial processing units, and also check whether the syntactic integration would be reflected at 1-second period, we conducted a frequency tagging analysis. By doing so, we constructed forty 15-second long trials for each participant by randomly selecting the neural response corresponding to the phrase condition and the sentence condition. Then the DFT was performed to extract the frequency neural response, decibel transformation was conducted based on the neural response at the baseline stage. Grand average was calculated to check the frequency domain characteristics. **Fig 8K** shows that there was a 1-Hz peak for both conditions. For checking the statistical significance, a paired-sample *t*-test, for both conditions, was conducted between the 1-Hz peak and the averaged frequency response around the it, with a window of five bins on each side. The 1-Hz peak was statistically significant for both the phrase condition (t (14) = 8.72, *p* < 4.9e-7 ***) and the sentence condition (t (14) = 8.46, *p* < 7.1e-7 ***). The results suggest that syntactic integration [24] happened at the 1-second period and that a 1-second duration normalization was effective. However, we can see that using a frequency tagging approach would make it difficult to separate the syntactic structures (t (14) = 0.63, *p* = 0.53).

**Fig 8l** shows the response spectrum around 4-Hz. A paired-sample *t*-test suggested that there was a strong 4-Hz response for both phrases (t (14) = 7.79, *p* < 1.8e-6 ***) and sentences (t (14) = 9.43, *p* < 1.9e-7 ***). The results suggest that syllables were the initial processing units for both phrases and sentences [24].

## Simulations based on a time-based binding mechanism for linguistic structure across neural ensembles

To perform simulations using the time-based binding mechanism from DORA, we constructed four-hundred 12-second long sequences for the input nodes (S-units, four nodes for each stimulus). As each phrase or sentence has four input nodes, our stimuli construction resulted in 100 phrase–sentence pairs. For each of the constructed sequences, an activation period that represents the occurrence of syllables was varied depending on its time order. For example, node S1 (**Fig 1A** or **1B**) represents the first syllable in a phrase or sentence. Therefore, we limited the range of its firing to the first 250 ms window in each one of the 12-second sequences. The same logic was applied to the remaining syllables, which is the firing window for S2, S3, and S4 was normalized to the second, third and fourth 250-ms time window in each second, respectively. To imitate the natural rhythm [42,46,47] and the nonisochronous feature of speech, the firing time of each syllable was manipulated to a random length in time, within the range of 140 ms to 230 ms (approximately 4 to 7 Hz), with a random starting point. Firing and resting in each time bin were initially represented by a binary classification (1 for firing, 0 for resting), then random white noise was stacked on. To normalize the amplitude of each sequence, a pointwise division with the maximum value of it was applied.

After constructing the input sequences, we conducted the simulation. To fit in the time-based binding feature across layers, one node and its sublayer units were linearly connected in three steps. The first step is the time-based binding via convolution, in which the activation of one node was a convolution between a unit step function scaled by its length and the summation of the activation of its sublayer units. This step reflects the critical property of time-based binding mechanism, which is an instantaneous activation of a unit is an average of the activation ahead of it within a defined window (see Fig 2 in Martin and Doumas [13]). The second step is low-pass filtering (FIR, nonphase changed), after acquiring a time bound signal, gaining SNR was applied. Low-pass filtering represents a refinement of the output and an increase in SNR. The last step is amplitude normalization, the sequence was finally standardized by a pointwise division with the maximum value of it to allow a cross-layer comparison. As nodes

in different layers reflect different levels of binding and SNR refinement, the window length of the kernel (step 1) and the high-cutoff threshold (step 2) was varied depending on the location of the parent nodes. Specifically, the length of the convolution kernel was 30-ms long when the parent node only has one sublayer unit and 250-ms long when it had two children. Note that the length of the window represents the effective range of the kernel as one side of it is all zeros (e.g., one instantaneous value only reflects the integration of the activation ahead of it). The high-cutoff for low-pass filtering was empirically defined as 50, 35 and 20-Hz for PO, RB and P-units, respectively, to reflect the hierarchy of SNR gaining.

Power was extracted using DFT, after which, power coupling (connectivity) was estimated between PO-units and S-units using Spearman (rank) correlation. Phase coherence was calculated by the average length of the phase reserved analytical signals (unit length) using PO-units. A 100-time randomization with thirty signals in each was conducted for each condition. Phase coupling (synchronization) was calculated between PO-units with its corresponding S-units, the same procedure was applied as the calculation of phase coherence, except that the phase series was extracted by cross-spectral density between the two types of nodes.

## Experimental procedure

Each trial started with a fixation cross at the screen center (500-ms in duration). Participants were asked to look at the fixation. Immediately after the fixation cross had disappeared, the participants heard a 1000-ms spoken stimulus, either a phrase or a sentence, followed by three-second silence; then the participants were asked to perform one of three types of task, indicated to them by an index (1, 2 or 3 showing at the screen center, 500-ms in duration). If the index was "1," they did a linguistic structure discrimination task (**type-one task**), in which they had to judge whether the spoken stimulus was a phrase or a sentence and indicate their judgment by pressing a button. If the index was "2," a picture would follow (200-ms in duration) and would be shown after a 1000-ms gap after the index number. Then participants would do a color matching task (**type-two task**), in which they had to judge whether the color described in the spoken stimulus matches the color of the shown picture and indicate their judgment by pressing a button. If the index is "3," they would experience the same procedure as the type 2 task, except for they would do an object matching task (**type-three task**), judging the matching of object between the spoken stimulus and the picture and indicating their respond by pressing a button. All responses were recorded via a parallel port response box, in which each one of the two buttons was represented as phrase/match and sentence/mismatch, respectively. Each response was followed by a silent interval of 3 to 4.5 seconds (jittered). The experimental procedure is depicted in the **Fig 9**.

Data collection was broken up into 5 blocks, with 48 trials in each block. Before the core data collection, several practice trials were conducted for each participant in order to make sure they had understood the task. Trials in each block were fully matched in across linguistic-structure (phrase or sentence) and task-type (type 1, type 2 or type 3). For example, half of the spoken stimuli were phrases and half were sentences (24 in each type), 6 types of combinations (8 trials for each type) were evenly distributed in each block ($8*2*3$), etc. Trial order was pseudo random throughout the whole experiment. The behavioral results indicated the task was relatively easy and no difference between phrase and sentences. All tasks combined accuracy for phrases and sentences were 97.9 ± 3% and 97.3 ± 3% ($p = 0.30$), respectively.

After the main experiment, a localizer task was performed, in which a tone beep (1-kHz, 50-ms in duration) was played 100 times (jitter 2 to 3 seconds) for each participant to localize the canonical auditory response (N1-P2 complex). The topographies for N1 and P2 are shown in **Fig 10**. The upper panel shows the averaged N1-P2 complex of all participants over the time

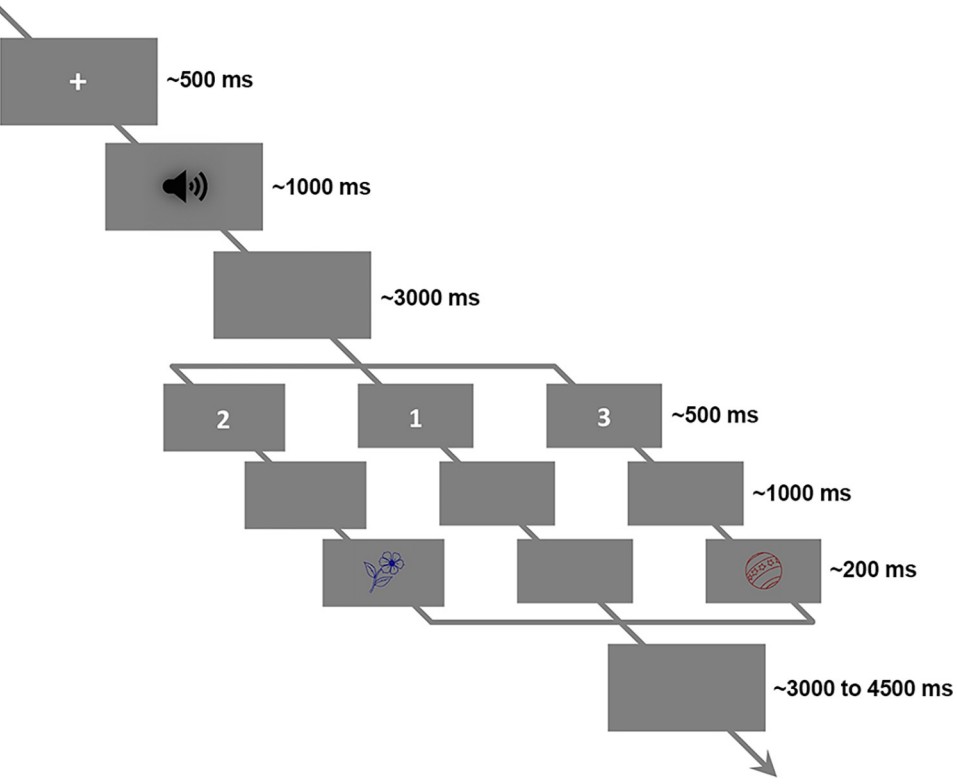

**Fig 9. Experimental procedure.** An illustration of the experimental procedure. Participants were asked to looking at the screen center, after hearing the speech stimulus they would do a task indexed by a number that showed on the screen. If the number was "1," they would judge whether the heard stimulus was a phrase or a sentence. If the number was "2," they would see a following picture then judge whether the color of the picture was the same as the color that described in the speech stimulus. If the number was "3," they would do an object matching task, in which they would judge whether the object of the picture was the same as the object that described in the speech stimulus. Trial type was pseudo randomly assigned throughout the whole experiment. The figure was originally created by Fan Bai, which permits unrestricted use and distribution. Available from: https://commons.wikimedia.org/wiki/File:Fig_9.tif.

bin from 90 to 110-ms for the N1 and 190 to 210 ms for the P2. The lower panel shows the N1-P2 complex after the Surface Laplacian [53,94,95], in which the effect of the volume conduction was attenuated. The topographies indicated that all participants had the canonical auditory response.

## EEG recording

We recorded EEG data using a 64-channel active sensors system of Brain Products (Brain Products GmbH, Gilching, Germany) in a sound dampened, electrically shielded room. Signals were digitized online at 1000 Hz, high-pass and low-pass at 0.01-Hz and 249-Hz, respectively. Two electrodes, AFz and FCz, were used as ground and reference. All electrodes were placed on the scalp based on the international 10–20 system and the impedance of each one was kept below 5-kΩ.

We presented stimuli using MATLAB 2019a (The MathWorks, Natick, Massachusettts, United States of America) with the Psychtoolbox-3 [96]. Auditory stimuli were played at 65-dB SPL and delivered through air tubes ear plugs (Etymotic ER-3C, Etymotic Research, Elk Grove Village, Illinois, United States of America). Event markers were sent via a parallel port for tagging the onset of the interested events (i.e., speech onset, task index onset, etc.).

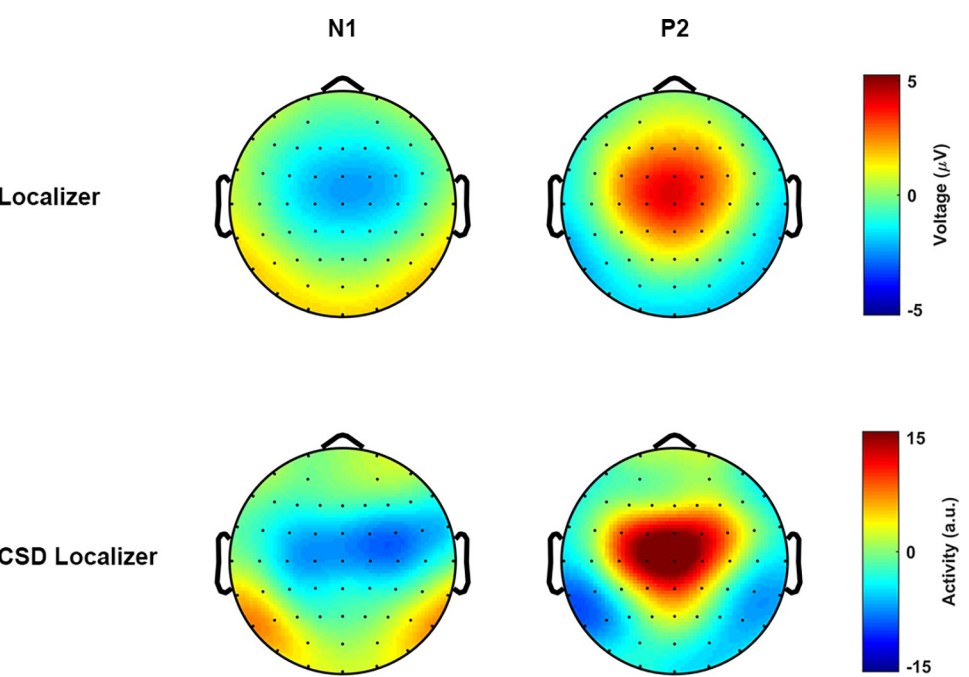

**Fig 10. Effect of volume conduction attenuation.** The topographical distribution of the canonical auditory N1-P2 complex. The upper panel shows the N1-P2 complex that was averaged over all participants ($N$ = 15). The lower panel shows the N1-P2 complex after Surface Laplacian (CSD), in which the effect of volume conduction was attenuated. CSD, current source density.

## EEG data preprocessing

We preprocessed EEG data via MATLAB using EEGLAB toolbox [97] and customized scripts. We first down-sampled the data to 256Hz, then high-pass filtered at 0.5-Hz (finite impulse response [FIR] filter; zero-phase lag), and cleaned by the time-sliding PCA [98,99]. We interpolated all detected bad channels with spherical interpolation. After transfer data to average reference, the online reference FCz was recovered and the line noise, 50-Hz and its harmonics, was removed.

Following the above-mentioned steps, we extracted epochs of 2-s preceding and 9-s after auditory stimulus onset. We conducted bad trials deletion and artifacts removal in the following two steps. First, we used independent component analysis (ICA) decomposing data into component space (number of components equals data rank). Then for each independent component, we used short-time Fourier transform (STFT) to convert each trial into power spectrum, in which we extracted a value which was calculated by the power summation between 15-Hz to 50-Hz. Then all the extracted values, one value per epoch, in each component formed a distribution. From this distribution, we transformed all the extracted values to z-score, the epochs with the value outside the range of plus and minus three standard deviations were deleted. Second, ICA was conducted again on the trial-rejected data for eye-related artifacts removal and sustained muscular activities elimination. Artifact components were identified and removed by an automatic classification algorithm [100]. All the preprocessing steps resulted in the removal of, on average, seven components (range 4 to 11) and 22 trials (including incorrect trials and trials with too slow response, range 10 to 30, 4% to 12.5%) per participant. Finally, volume conduction was attenuated by applying the Surface Laplacian [53,94,95].

## EEG data analysis

**Time frequency decomposition.** We convolved the single-trial time series with a family of complex wavelets (1 to 50-Hz in 70 logarithmically spaced steps), temporal and spectral resolution optimization was conducted by changing the cycle from 3 to 30 in logarithmical steps.

We calculated phase coherence by intertrial phase clustering (ITPC) [53, 101]. At each time–frequency bin, the wavelet coefficients of all trials were divided by their corresponding magnitude and averaged across trials. The magnitude of the averaged complex output was represented as phase coherence (ITPC).

The induced neural response (power) was extracted from the analytical output at each time–frequency bin by taking the summation of the squared wavelet coefficients. Decibel transformation was performed separately at each frequency, in which the average power at the duration from 800-ms to 200-ms before the audio onset was used as the baseline.

**Phase connectivity.** We first decomposed trials in each condition via wavelet convolution (same parameters as the time–frequency decomposition). Then, we calculated the Cross Spectral Density (CSD) for each sensor-pair at each frequency-time-trial bin. We calculated phase connectivity over the sensor space by ISPC [53,57,58,101], in which we divided the complex coefficients by the corresponding amplitude at each frequency-time-trial bin then computed averages across all trials. The amplitude of the averaged complex vector was represented as phase connectivity between sensors (ISPC). After the above-mentioned steps, the phase connectivity at each time–frequency bin was represented as an all-sensor to all-sensor matrix, in our case, 65 sensors $*$ 65 sensors.

To transform the connectivity matrix to connectivity degree at each time–frequency bin, a statistical thresholding method was conducted. More specifically, at each time–frequency bin, we formed a distribution by pooling together all the connectivity values from both conditions, then defined the threshold as the value which is half a standard deviation above the median. We then binarized the connectivity matrices for both conditions by applying this threshold at each bin. The connectivity degree at each time–frequency bin was then represented as the count of the number of the connectivity values that above this threshold. Finally, we normalized the connectivity level at each time–frequency bin to percentage change relative to the baseline which was calculated as the average connectivity degree at the duration from 800 ms to 200 ms before the audio onset.

**PAC.** Since low-frequency phase and high-frequency amplitude may show coupling during speech processing [49], we defined the frequency range for the phase time series from 1 to 16-Hz in a linear step of 1.5-Hz, and the frequency range for the amplitude time series from 8-Hz to 50-Hz in 12 logarithmical steps. Then, we performed the wavelet convolution to extract the analytic signals, in which we extracted the phase time series and amplitude time series at the specified frequency range from 50-ms before to 1500-ms after the audio onset. At each phase–amplitude bin, we constructed a complex time series that held the phase angle of the phase time series and weighted by the magnitude of the amplitude time series. We calculated the PAC at each bin by extracting the magnitude of the average of all the vectors in the complex time series [53,78]. Since the variation of the amplitude response, a Z-score normalization was also performed for each phase–amplitude bin. More specifically, after calculating the real PAC value using the raw complex time series, the random PAC value were computed 1000 times using the re-constructed complex time series, which were built by temporally shifting the amplitude time series with a random temporal offset. These 1000 random PAC values formed a reference distribution for each phase–amplitude bin. Then the z-score of the real PAC value in this distribution was represented as the index of the phase–amplitude coupling, PAC-Z.

**Power connectivity.**   After time–frequency decomposition, we extracted induced power at each channel-time–frequency-trial bin. For each condition, we calculated power connectivity for each sensor-pair at each time–frequency bin as the Rank correlation between the power response of all trials in one sensor and the power response of all trials in the other sensor [53,102,103]. The power connectivity calculation resulted in an all-sensors to all-sensors (65*65 in our case) representation at each time–frequency bin for each condition.

To transfer the power connectivity at each time–frequency bin to the power connectivity degree, we applied a statistical thresholding method. More specifically, at each time–frequency bin, we formed a distribution by pooling together all the connectivity values from both conditions, then defined the threshold as the value which is half a standard deviation above the median. We then binarized the connectivity matrix at each bin for each condition by applying the corresponding threshold. The connectivity degree at each time–frequency bin was represented as the count of the number of the connectivity values that above this threshold. Finally, we transferred the connectivity level at each time–frequency bin as a percentage change relative to the connectivity level of the baseline which was calculated as the average connectivity level in the duration from 800 ms to 200 ms before the audio onset.

**STRF.**   The STRF is a linear kernel which convolves with the specified features of the speech signal to estimate the neural response in time. It can be interpreted as a linear filter which transforms the stimulus feature (SF) to the neural response [75,104].

In our study, the SF was defined as the spectrogram, which was obtained by filtering the speech stimulus into 16 logarithmically-spaced frequency bands between 0.05 and 8 kHz for mimicking frequency decomposition by the brain [105]. The temporal envelope (energy profile) for each frequency band was then extracted via the Hilbert transform.

To construct the stimulus-response pairs, we first applied a linear ramp to both sides of each trial corresponded neural response (10% at each side) to attenuate the abrupt onset and offset. Then, we matched each 1-second neural response with the corresponding SF.

In order to optimize the estimation of the STRF, a randomization procedure was applied to create new data structure. We first randomly selected 80% of all unique speech stimuli, then the stimulus-response pairs that corresponded to the selected speech stimuli were extracted as the seed data to construct the training dataset for performing the cross-validation. We constructed thirty-five 10-second long stimulus-response pairs, in which each one of them was a concatenation of the randomly selected ten 1-second stimulus-response pair (Bootstrapping).

The STRF was estimated using ridge regression with leave-one-out cross-validation. Since ridge regression weights the diagonal elements of the covariance matrix of the SF with a parameter lambda [104,106], we, predefined the range of lambda to 10 values from 6 to 100 in linear steps. We used the extracted training dataset (thirty-five 10-second long stimulus-response pairs) to conduct the cross validation for optimizing the STRF. The Pearson correlation between each real neural response and each predicted response was calculated. The average of all the coefficients of the Pearson correlation (over all sensors and all trials) was defined as the performance of the STRF. The model with the lambda parameter which gave the best performance was used as the optimized STRF.

Since previous research has shown that slow timescale (low-frequency) neural responses reliably reflects the neural representation of the acoustic features in speech [72,73], we initially checked whether the STRF for different frequency bands faithfully reflected the encoding of the acoustic features. To do so, we first filtered the neural response corresponding to each trial into five canonical frequency bands, which were delta (<4-Hz), theta (4 to 7-Hz), alpha (8 to 13-Hz), beta (14 to 30-Hz), and low-gamma (31 to 50-Hz), respectively. Then, the STRF for each condition at each frequency band was estimated using the procedure that mentioned above. In order to check the performance of each estimated STRF, the stimulus-response pairs

(the unseen pairs for each STRF) that corresponded to the remaining 20% of the speech stimuli were extracted as the seed data for constructing the testing dataset. We extracted five 4-second long stimulus-response pairs in the testing data for each STRF, each one of them was a concatenation of randomly selected four 1-second long pairs.

The real performance of each STRF was calculated by using the frequency and condition matched stimulus-response pairs. The random performance was calculated 1000 times using the pairs which were constructed by combining the stimulus with a randomly selected neural response.

For fitting the low-frequency STRF (<9-Hz), the same procedure was conducted. The performance of the averaged STRF in each condition was computed using the averaged Pearson correlation between the real neural response and the predicted response across sensors.

We extracted the TRF and the SRF for each participant at each condition by averaging the STRF over the frequencies from 0.1-kHz to 800-kHz and by averaging the STRF over the time from 0 to 400-s, respectively.

All the calculations in this section were conducted using customized scripts, the scripts of EEGLAB toolbox [97], and the Multivariate Temporal Response Function Toolbox [104].

**Statistical analysis.** In addition to using parametric statistical methods, we applied a cluster-based nonparametric permutation test. This method deals with the multiple-comparisons problem and at the same time considers the dependencies (temporal, spatial, and spectral adjacency) in the data. For all types of analysis that followed this inference method, we initially averaged the subject-level data over trials and for each single sample, i.e., a time–frequency-channel point, and performed a dependent $t$-test. We selected all samples for which the t-value exceeded an a priori threshold, $p < 0.05$, two-sided, and subsequently clustered these on the basis of spatial and temporal–spectral adjacency. The sum of the t-values within a cluster was used as cluster-level statistic. The cluster with the maximum sum was subsequently used as the test statistic. By randomizing the data across the conditions and recalculating the test statistic 1000 times, we obtained a reference distribution of the maximum cluster t-values. This distribution was used to evaluate the statistical distribution of the actual data. This statistical method was implemented using the Fieldtrip toolbox [107,108].

## Discussion

In this study, we investigated the neural responses to linguistic structures by minimizing their differences in acoustic–energetic/temporal–spectral profiles and semantic components. We investigated which dimensions of neural activity discriminated between the syntactic structure of phrases and sentences, as cued by morphemes, and used a series of analysis techniques to better describe the dimensions of neural readout that were sensitive to the distinctive linguistic structures between phrases and sentences. We asked whether phrases and sentences had different effects on functional connectivity, and found, first, that while phrases and sentences recruit similar functional networks, the engagement of those networks scaled with linguistic structure (viz., presence of inflectional morphemes, number and type of syntactic constituents, and the relations between them). Sentences showed more phase coherence and power connectivity compared to phrases. This connectivity pattern suggests that phrases and sentences differentially impacted the distribution and intensity of the neural networks involved in speech and language comprehension. Second, we found that phase–amplitude coupling between theta and gamma, which has been implicated in speech processing, is not sensitive to structural differences in spoken language. Third, we found that activity in the alpha band was sensitive to linguistic structure. Last, by modeling acoustic fluctuations in the stimulus and brain response with STRFs, we found that phrases and sentences differentially relied on the encoding of

acoustic features in the brain, and that sentences were more abstracted away from acoustic dynamics in the brain response. We also demonstrated that a time-based binding mechanism, as instantiated in the symbolic-connectionist model DORA, predicts and exhibits the power and phase coupling patterns observed in our EEG data. In the following sections we give more detail about our findings and discuss potential interpretations of them.

## Phase coherence

Consistent with previous research [42,44,45,109], our phase synchronization analysis detected low-frequency phase coherence during speech comprehension. Moreover, phase coherence distinguished between phrases and sentences, yielding a cluster between approximately 450 and approximately 900-ms after audio onset and from approximately 2-Hz to approximately 8-Hz in frequency, which was most pronounced over central electrodes. These results therefore suggest that syntactic structure may be encoded by low-frequency phase coherence, through systematic organization of activity in neural networks, in particular their temporal dynamics. Our results are consistent with the notion of *phase sets* in computational models of structured representations that exploit oscillatory dynamics. Phase sets are representational groupings that are formed by treating distributed patterns of activation as a set when units are in (or out) of phase with one another across the network [11–13,110]. They are key to the representation of structure in artificial neural network models, as seen in the simulated predictions reported here.

In our simulation of the time-based binding mechanism, we demonstrated that there is a difference in phase synchronization between conditions between 250 and 1000-ms, driven directly by how the model represents syntactic structures using time-based binding; this window overlaps with what was observed in the EEG data (450 to 900-ms). Note that an observational conundrum exists for many established brain responses in terms of timing between observed response and cognitive process. For example, the P600 effect in syntactic processing is viewed as a late effect in relation to other event-related brain potentials, while it is known that differences in the brain arise much earlier during syntactic processing—behavioral responses during the detection of syntactic violations occur much earlier than the P600, and earlier than the detection of semantic violations (e.g., [111]). Given this pattern, neural responses can be seen as the upper limit on temporal processing in the brain, rather than veridical one-to-one signals reflecting real-time cognitive processing, simply due to measurement, estimation, and signal-detection issues. In sum, the results of the simulation demonstrated that the degree of phase synchronization between layers of nodes varies with number and type of syntactic structures, as well as the relations between them.

## Phase connectivity

Phrases and sentences also yielded differences in phase connectivity. In the predefined time and frequency range of interest, the statistical comparison indicated a difference corresponding to a cluster approximately from approximately 800 to approximately 1600-ms after auditory offset, occurring at a very low-frequency range (approximately <2-Hz) that was most pronounced over the posterior right hemisphere. Phrases and sentences thus differentially impact the temporal synchronization of neural responses.

Several aspects of the results are noteworthy. First of all, the relatively late effect suggests that the effect on temporal synchronization occurs after the initial presentation of the speech stimulus. In our experiment, participants were pseudorandomly presented a task prompt for one of three possible tasks (i.e., color discrimination, object discrimination, phrase or linguistic structure discrimination), which asked them to determine either "semantic" (object or

color information) or "syntactic" information (viz., whether the stimulus was a phrase or sentence) from the speech stimulus. Because of the pseudorandom order of the task trials, participants had to pay close attention the stimuli and continue to represent each stimulus after hearing it, namely until they received the task prompt. The tasks also insured that participants could not select a single dimension of the stimulus for processing. Similarly, because we used an object and a color task, participants had to distribute their attention evenly across the adjectives and nouns, mitigating word order differences between structures. In light of these controls and task demands, we consider it unlikely that the observed phase connectivity effects reflect mere differences in attention to phrases or sentences. Rather, we attribute the observed effects to the syntactic differences between them.

Second, the low-frequency range (<2-Hz) of the observed effect is consistent with previous research [24–26,29,39]. In Ding and colleagues [24], the cortical response was modulated by the timing of the occurrence of linguistic structure; low-frequency neural responses (1 to 2-Hz) were found to track the highest-level linguistic structures (viz., phrases and sentences) in their stimuli. Here, we extended their work to ask whether the 1-Hz response could be decomposed to reflect separate syntactic structures (e.g., phrases versus sentences), and we identified the role of phase in discriminating between these structures. In our study, all speech stimuli lasted 1-second, and except for the presence of morphemic and syntactic structure, the stimuli were normalized to be highly similar. Our pattern of results therefore suggests that functional connectivity, as reflected in the temporal synchronization of the induced neural response, distinguishes between phrases and sentences.

Last, phrases and sentences differed most strongly over the posterior right hemisphere. We hesitate to infer computational process based on temporal or regional effects alone, especially for phase synchronization. However, our results are broadly consistent with previous research on syntactic processing, although they cannot be unequivocally attributed to syntactic processing alone. Neurophysiological research suggests the involvement of the right hemisphere in the extraction of slow timescale information [112–115]. The P600, an ERP component often associated with syntactic processing, has a robust topographical distribution with right hemisphere dominance between 600 and 1000-ms poststimulus onset [116–121]. Thus, in principle, the right-posterior distribution of the phase connectivity effects is consistent with existing putatively syntactic effects, but we refrain further interpretation of the computational nature of this effect and its underlying neural sources based on our EEG data.

## PAC

We observed PAC during speech comprehension, as a low-frequency phase response (approximately 4 to 10-Hz) entrained with high-frequency amplitude (approximately 15 to 40-Hz). This effect appeared over a largely bilateral, central area. The bilateral central topographical distribution has been implicated in sensory-motor integration [122–128], which is consistent with the proposal from Giraud and Poeppel [49] that PAC reflects an early step in speech encoding involving sensory-motor alignment between the auditory and articulatory systems. Crucially, however, this effect did not discriminate between phrases and sentences. Although a null result, the observed pattern is compatible with the generalized model for speech perception [49]. This early step is presumably similar for phrases and sentences and perhaps for any type of structure above the syllable level.

### Induced alpha power

Induced alpha-band power distinguished phrases and sentences, and this effect was most pronounced over the left hemisphere. This pattern suggests involvement of alpha-band

oscillations in syntactic structure processing. Although alpha-band activity is often associated with attentional or verbal working memory [60–67], we do not consider this a very plausible alternative explanation for our results. For example, it is not clear why phrases and sentences would differ in their attentional or working memory demands. In addition, we employed an experimental task to ensure similar attention to phrases and sentences, and phrases and sentences were associated with similar behavioral performance in each task (with a caveat that performance was at ceiling and may therefore not pick up on small differences between conditions).

We do not claim that that all speech-elicited alpha-band effects reflect syntactic processing. Some observed effects may well reflect perceptual processing during speech comprehension (e.g., [71]), especially in experiments designed to manipulate perceptual processing, such as speech-in-noise manipulations. The neural response in a given band, e.g., the alpha band, need not reflect only one particular perceptual process. Likewise, the fact that the alpha-band neural response could reflect lower-level perceptual processes or working memory load in certain contexts does not necessarily rule out its role in higher-level linguistic information representation, e.g., syntactic processing.

## Power connectivity

Phrases and sentences elicit differences in induced power connectivity in alpha-band activity (approximately 7.5 to 13.5-Hz). Phrases showed more inhibition in power connectivity than sentences; in other words, sentences showed a stronger connectivity degree over sensor space in the alpha band than phrases. Several aspects of these results are noteworthy. First, we observed this effect from approximately 100-ms until approximately 2200-ms after stimulus onset, which suggests the effect in functional connectivity persisted and outlasted the observed effects in induced alpha power, which we observed from approximately 350-ms to approximately 1000-ms after auditory onset i.e., (during the listening stage). The extended nature of the functional connectivity effect could reflect the continuing integration and representation of syntactic and semantic components.

Second, alongside differences between phrases and sentences in power connectivity, we also extracted the sensor connectivity pattern (over an ROI ranging from 100-ms to 2200-ms in time and 7.5 to 13.5-Hz in frequency). Whereas phrase and sentences showed similar functional connectivity in the intensity of the neural response, sentences showed stronger interregion (sensor) connectivity than phrases. By design, in our stimuli sentences had more constituents than phrases did. If local network activity is more organized or coherent as a function of linguistic structure, then the difference observed here could reflect the encoding of additional constituents in sentences compared to phrases.

Lastly, phrases elicited stronger inhibition of induced power connectivity than sentences did. This indicates weaker cooperation between brain regions, in other words, regions showed more independence in neural response. In contrast, interregion connectivity was stronger for the sentence condition than the phrase condition, which suggested a higher-level of intensity of connectivity between brain regions for sentences that distinguished them from phrases.

In sum, phrase and sentences elicited robust differences in induced power connectivity. A similar functional connectivity pattern was deployed for representing phrases and sentences, but the intensity of the connectivity was stronger for sentences than phrases. This finding is consistent with the prediction that low-frequency power and network organization should increase as linguistic structure increases. Our stimuli were designed to allow the measurement of differences in neural dynamics between phrases and sentences, and as such differed in the number and type of linguistic constituents that were perceived. Beyond the number and type

of constituents, the phrase and sentence structures also differ in the relations between constituents, or in the linguistic notion of hierarchy. But given the co-extension of number, type, and relation, our stimuli and design do not yet allow us to determine if, nor how, these variables alone might affect structural encoding in neural dynamics.

## STRFs

We performed STRF analysis to investigate whether phrases and sentences are encoded differently. First, consistent with previous research [72,73], only low-frequency (<9-Hz) neural responses robustly reflected the phase-locked encoding of the acoustic features. Second, we observed a bilateral representation of the slow temporal modulations of speech, in particular at posterior sensors. The posterior effects have consistently been found in response to syntactic integration [116–121]. The low-frequency neural response that models the phase-locked encoding of acoustic features can capture structural differences between phrases and sentences, even without explicitly using a hand-coded annotation on the syntactic level to reconstruct the data. Third, and most importantly, we explored these patterns further in both the temporal and spectral dimension, by decomposing the STRF into the TRF and the SRF. Both TRF and SRF suggested a different encoding mechanism between phrases and sentences. More specifically, the TRF results showed that the brain transduces the speech stimulus into the low-frequency neural response via an encoding mechanism with two peaks in time (at approximately 100-ms and approximately 300-ms). The two peaks reflect the instantaneous low-frequency neural response that is predominantly driven by the encoding of acoustic features that were presented approximately 100-ms and 300-ms ago. In the two-time windows that centered at approximately 100-ms and 300-ms, respectively, phrases and sentences showed a different dependency on the acoustic features in both latency and intensity.

When we only consider intensity (approximately 100-ms time window), sentences depended on acoustic features less strongly than phrases. This result is consistent with the idea that sentence representations are more abstracted away from the physical input because they contain more linguistic structural units (i.e., constituents) that are not vertically present in the physical or sensory stimulus. Consistent with previous research, we found that the instantaneous neural response was strongly driven by the encoding of the acoustic features presented approximately 100-ms ago [30,72–75,129–132].

When we only consider the latency (approximately 100-ms time window), and only the right hemisphere, the low-frequency neural response to sentences was predominantly driven by the acoustic features that appeared earlier in time than the acoustic features that drove the neural response to phrases. Our results imply that the brain distinguishes syntactically different linguistic structures by how its responses are driven by the acoustic features that appeared approximately 100-ms ago. More importantly, over the right hemisphere, our findings suggest that the low-frequency neural response to sentences reflected the encoding of the acoustic features that appeared earlier in time than the acoustic features that drove the neural response to phrases. This could be evidence that the right hemisphere is dominant in extracting the slow timescale information of speech that is relevant for, or even shapes, higher-level linguistic structure processing, e.g., syntactic structure building [72,73,113]. It is noteworthy to see that the distribution in time and space of these patterns is consistent with the idea that the brain is extracting information from the sensory input at different timescales and that this process is, in turn, is reflected in the degree of departure (in terms of informational similarity) of the neural response from physical features of the sensory input.

At approximately 300 ms, when we only consider intensity, the low-frequency neural response to phrases is more strongly dependent on the acoustic features than the neural response to sentences is, showing again that sentences are more abstracted from sensory representations. However, the comparison of the TRF indicated that the brain exploited a different encoding mechanism at the right hemisphere between the phrases and the sentences. More concretely, the low-frequency neural response to phrases showed a stronger dependency on acoustic features than the neural response to sentences did over the right hemisphere, but not over the left hemisphere. The results, first, suggested that the instantaneous low-frequency neural response reflects the encoding of acoustic features that were presented approximately 300-ms ago, in both conditions. Then, it reflected that, only in the right hemisphere, the low-frequency neural response of the phrases more strongly depended on the acoustic features from approximately 300-ms ago when compared to the neural response to sentences. This finding reminds us of the results of the phase connectivity analysis, in which the phase connectivity degree also showed a different pattern between the phrases and the sentences over the posterior right hemisphere; which is consistent with previous research that has demonstrated a role for the right hemisphere in processing slow modulations in speech, such as prosody [44,72,73,112–114,133]. Consistent with these findings, our results further indicated that the brain can separate syntactically different linguistic structures via differential reliance by the right hemisphere on representations of acoustic features that appeared at approximately 300 ms ago. That sentence representations were more abstract and less driven by the acoustics in the left hemisphere is consistent with contemporary neurobiological models of sentence processing [5,6], although we cannot attribute the differences we observed to a single aspect of linguistic structural descriptions (i.e., to number, type, relations between constituents, or whether constituents form a "saturated" sentence or not).

The SRF results indicated that the brain can begin to separate phrases and sentences via differential reliance on the encoding of acoustic features from roughly the first formant, and in a phase-locked manner. More specifically, in the range of the first formant, the low-frequency neural response reflected a stronger dependency on the acoustic features for phrases than for sentences. Unlike consonants, the intensity of vowels is well reflected at the first formant (<1 kHz) [80–83]. Although the overall physical intensity of the speech stimulus of the phrases was not different from the sentences, the endogenous neural response to the stimulus contains information that does discriminate between syntactic structures. Given that the stimuli were not physically different, this pattern of results strongly suggests that the brain is "adding" information, for example, by actively selecting and representing linguistic structures that are cued by the physical input and its sensory correlate. For example, the brain could be adding phonemic information, e.g., via a top-down mechanism; in certain situations, and languages, even a single vowel can cue differential syntactic structure. In fact, in our stimuli, the schwa carries agreement information that indicates the phrasal relationship between "red" (*rode)* and "vase" (vaas) in the phrase "de rode vaas." Our results, which feature both dependence on, but also departure from, the acoustic signal, are consistent with previous findings that have shown low-frequency cortical entrainment to speech and have argued that can reflect phoneme-level processing [26,75,134,135]. We extend these findings by showing that when lower-level variables in the stimuli are modeled, the brain response to different syntactic structures can be decomposed even without the addition of higher-level linguistic annotations in our TRF models, and that this result indicates that the degree of departure from the physical stimulus increases as abstract structure accrues.

## Summary

In this study, we found a neural differentiation between spoken phrases and sentences that were physically and semantically similar. Moreover, we found that this differentiation was

captured in several readouts, or dimensions of brain activity (viz., phase synchronization, functional connectivity in phase and induced power). By modeling the phase-locked encoding of the acoustic features, we discovered that the brain can represent the syntactic difference between phrases and sentences in the low-frequency neural response, but that the more structured a stimulus is, the more it departs from the acoustically-driven neural response to the stimulus, even when the physicality of the stimulus is held constant. We demonstrated that a time-based binding mechanism from the symbolic-connectionist model DORA [11–13,41] can produce the power and phase coherence patterns we found in our EEG data. These predictions, confirmed through simulation, in combination with our observed data, suggest that information processing across distributed ensembles in a neural system can be adequately described by such a time-based binding mechanism. Across all our results, we provide a comprehensive electroencephalographic picture of how the brain separates linguistic structures within its representational repertoire. However, further research is needed to explore the relationship between these different neural readouts that indexed syntactic differences, e.g., how the induced neural response in the alpha band interacts with phase coherence in the low-frequency (<8 Hz), and how these separation effects are represented at the neural source level.

## Supporting information

**S1 Appendix. All the phrase–sentence pairs that were used in the experiment.** (PDF)

## Acknowledgments

The authors thank Mante S. Nieuwland and Sanne ten Oever for comments on an earlier version of this work, Shuang Bi for selecting stimuli and arranging figures, Cas W. Coopmans for guidance as to the visual representation of linguistic constituents, Hugo Weissbart for suggestions regarding acoustic normalization, and Karthikeya Kaushik for consultation regarding the simulations.

## Author Contributions

**Conceptualization:** Fan Bai, Andrea E. Martin.

**Data curation:** Fan Bai.

**Formal analysis:** Fan Bai, Andrea E. Martin.

**Funding acquisition:** Antje S. Meyer, Andrea E. Martin.

**Investigation:** Fan Bai.

**Methodology:** Fan Bai, Andrea E. Martin.

**Project administration:** Andrea E. Martin.

**Resources:** Fan Bai.

**Software:** Fan Bai.

**Supervision:** Antje S. Meyer, Andrea E. Martin.

**Validation:** Fan Bai, Andrea E. Martin.

**Visualization:** Fan Bai, Andrea E. Martin.

**Writing – original draft:** Fan Bai, Andrea E. Martin.

**Writing – review & editing:** Antje S. Meyer, Andrea E. Martin.

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
