## [Editor Report · Decision Letter 0]

15 Feb 2022

Dear Andrea, 

Thank you for submitting your revised manuscript entitled "Neural encoding of phrases and sentences in spoken language comprehension" for consideration as a Research Article by PLOS Biology.

Your new version has now been evaluated by the PLOS Biology editorial staff, as well as by the original academic editor, and I am writing to let you know that we would like to send your submission out for external peer review. Please accept my apologies for the delay in sending this decision to you.

Once your full submission is complete, your paper will undergo a series of checks in preparation for peer review. Once your manuscript has passed the checks it will be sent out for review. To provide the metadata for your submission, please Login to Editorial Manager (https://www.editorialmanager.com/pbiology) within two working days, i.e. by Feb 17 2022 11:59PM.

If your manuscript has been previously reviewed at another journal, PLOS Biology is willing to work with those reviews in order to avoid re-starting the process. Submission of the previous reviews is entirely optional and our ability to use them effectively will depend on the willingness of the previous journal to confirm the content of the reports and share the reviewer identities. Please note that we reserve the right to invite additional reviewers if we consider that additional/independent reviewers are needed, although we aim to avoid this as far as possible. In our experience, working with previous reviews does save time. 

If you would like to send previous reviewer reports to us, please email me at ggasque@plos.org to let me know, including the name of the previous journal and the manuscript ID the study was given, as well as attaching a point-by-point response to reviewers that details how you have or plan to address the reviewers' concerns. 

Given the disruptions resulting from the ongoing COVID-19 pandemic, please expect some delays in the editorial process. We apologise in advance for any inconvenience caused and will do our best to minimize impact as far as possible.

Kind regards,

Gabriel

Gabriel Gasque

Senior Editor

PLOS Biology

ggasque@plos.org

---

## [Decision Letter · Decision Letter 1]

15 Mar 2022

Dear Dr Martin,

Thank you very much for submitting a revised version of your manuscript "Neural encoding of phrases and sentences in spoken language comprehension" for consideration as a Research Article at PLOS Biology. I'm now handling your paper as my colleague Dr Gabriel Gasque has sadly left PLOS. This revised version of your manuscript has been evaluated by the PLOS Biology editors, the Academic Editor and two of the original reviewers.

You'll see that while reviewer #2 is now mostly satisfied, reviewer #1 raises some continuing concerns that will need to be addressed. I asked the Academic Editor about this reviewer's requests, and s/he said "I think this paper still needs substantive revisions -- although I think that if they are done well, the paper will be in good shape and be acceptable. The major comment made by reviewer #1 really does need to be addressed for the argument to be compelling."

In light of the reviews (below), we will not be able to accept the current version of the manuscript, but we would welcome re-submission of a much-revised version that takes into account the reviewers' comments. We cannot make any decision about publication until we have seen the revised manuscript and your response to the reviewers' comments. Your revised manuscript is also likely to be sent for further evaluation by the reviewers.

We expect to receive your revised manuscript within 3 months. 

**IMPORTANT - SUBMITTING YOUR REVISION**

*Re-submission Checklist*

*Published Peer Review*

*PLOS Data Policy*

*Blot and Gel Data Policy*

Sincerely,

Roli Roberts

Roland G Roberts PhD

Senior Editor

PLOS Biology

rroberts@plos.org

REVIEWERS' COMMENTS:

Reviewer #1:

SUMMARY

I thank the authors for taking the time and care to address my comments. While my concerns regarding the acoustic matching of the two primary conditions are alleviated, I still have hesitation regarding the overall claims and interpretations of the work. In their response to my review, the authors state "Our experiment tests the predictions of a computational model published in this journal (Martin & Doumas, 2017), which proposed a mechanism for building syntactic structure, time-based binding." For this to be true, I think ideally the link between the current empirical results, and the explicit predictions for the prior computational model need to be made explicit through simulation. 

MAJOR

(1) Link to Martin & Doumas (2017)

This work is being used to provide empirical evidence for the claims put forward in a previous paper published in the same journal. However, to be used as such, I feel like the appropriate approach would be to take the DORA model used in the previous paper, simulate the responses to the same stimuli used in this study, and compare to the obtained EEG signals. Without a direct comparison, and without a detailed explanation of what the previously detailed "mechanism" is that is being supported here, it is very hard for the reader to validate the results. 

The previous paper found that the DORA model was also able to capture the frequency tagging result of Ding et al., 2016. But to make claims regarding the binding of information through aligned responses in time, using non-isochronous stimuli, I think that we would need to see that the DORA model predicts phase alignment during the same time periods observed in the EEG responses.

Without this, I think the next best thing is to provide a schematic of the proposed mechanism, so that the reader can understand the concrete predictions without having to refer back to the previously published study.

(2) Time alignment 

The authors present their results time-locked to sentence onset. But the listener will distinguish sentence vs. Phrasal structure at later (and potentially multiple) points within the stimulus. In order to link the current results to a binding mechanism through phase coupling, I feel that the timing of the effects need to be directly linked to the syntactic structure available to the system at that moment in time.

MINOR

- in some places I feel like the writing is unnecessarily complicated. For example, by "the acoustic and physical dynamics of speech do not injectively mark the linguistic structure and meaning that we perceive" do you mean that linguistic structure cannot be directly read out from the acoustic signal? There are a few instances where I think the language could be simplified a bit.

- I found it a bit distracting the sentence "speech contains an abundance of acoustic features in both time and frequency" … speech IS an acoustic signal, which can be defined, if you choose, relative to the dimensions of time and frequency. Do you mean that it is particularly variable along these dimensions? It is not just that speech contains acoustic features, it cannot be divorced from those features.

- Lines 1039:1042 seem circular / repetitive to me

- Typo "organisation" on line 1182

Reviewer #2:

[identifies himself as William Matchin]

I am mostly satisfied with the responses and revisions that the authors have provided to my concerns on the original submission. However I still have some minor concerns.

In the response document the authors note that they have provided a gloss to figure 1, but I do not see this - all I see is the Dutch syllable, without information regarding the linguistic function of these elements. I feel that some of this detail should also be discussed earlier in the paper - that it's not just the hierarchical arrangement that differs between the sentence and phrase conditions, but also the addition of a function word in the sentence condition and an agreement morpheme in the phrase condition.

Some general comments on language - the manuscript is dense and often difficult to understand for researchers less familiar with the terminology and background. For example, line 122-123, "the relative timing of neural ensemble firing is taken as an informational degree of freedom". This is hard to unpack. I also had a hard time parsing the following sentence, lines 123-125. "distributed representations that fire together closely in time, pass activation forward to receiver ensembles, such as words, are encoded together by those receiver ensembles…". It's not clear to me if there is some sort of grammatical issue here or the sentence is just hard to understand. I believe that more plain language and concrete examples would go a long way towards making these ideas more intelligible to the reader who is less familiar with these concepts, which is particularly important given the general scope of PLoS Biology.

It is hard for me to interpret the claims that the physical stimuli between the phrase and sentence conditions are "highly similar", since it is not clear what the relevant baseline to me is. For example, if we take random pairs of sentences in the present stimulus set, what would the physical similarity be? I think any claims of "similar pattern", "highly similar", etc., need to be evaluated against some relevant baseline. I'm not 100% sure what that baseline should be, but of course there are going to be significant correlations between two speech stimuli of similar length. Case in point, figures 1c and 1d claim to show that the temporal-spectral pattern is similar, but they definitely have noticeable differences, for example the much stronger high frequency energy in 1d at ~220-250 ms that is missing in 1c, etc. Figure 1e makes is hard to examine the true differences in conditions because the lines are thick and squished together. On this point, it seems to me that while showing significant acoustic similarity between the conditions is relevant, this does not show that there are no differences, and that these differences might contribute to the results.

I feel that this should be acknowledged a bit more clearly in the manuscript - that it's of course impossible to rule out acoustic differences contributing to these effects, but that it's unlikely that this is happening for various reasons.

I found a number of typos throughout the manuscript; it would probably be best to search exhaustively for such errors.

line 126 - "extended in to" -> "extended to/into"

Line 707 - synatctic -> syntactic

figure 1 f: internsity -> intensity

Figure 9 caption: volumn -> volume

line 313: acoustsics -> acoustics

Figure 4 caption: mechnism -> mechanism

Line 504: boarder -> border

---

## [Decision Letter · Decision Letter 2]

26 May 2022

Dear Dr Martin,

Thank you for your patience while we considered your revised manuscript "Neural encoding of phrases and sentences in spoken language comprehension" for publication as a Research Article at PLOS Biology. This revised version of your manuscript has been evaluated by the PLOS Biology editors, the Academic Editor and one of the original reviewers.

Based on the reviewer feedback and our Academic Editor's assessment of your revision, we would like to move towards acceptance of this manuscript for publication. At this stage we simply require that you satisfactorily address the remaining editorial issues, and that you address our data and other policy-related requests (outlined at the bottom of this email).

***Editorial requests:

Suggested title change to: Neural dynamics differentially encode phrases and sentences during spoken language comprehension

Rewrite of the Abstract to ensure broad accessibility: 

We feel that the content of your study has potential broad appeal outside of the cognitive neurosciences, to both other scientists and even non-scientists. To ensure that the study is more broadly accessed, we would like you revise the abstract to ensure that the work can be understood by people outside the field. We recommend that you have a colleague in another field have a look over a revised version of your abstract to gauge its accessibility. 

We expect to receive your revised manuscript within two weeks. 

*Published Peer Review History*

*Press*

Sincerely,

Kris

Kris Dickson, Ph.D. (she/her),

Neurosciences Senior Editor/Section Manager,

kdickson@plos.org,

PLOS Biology

DATA POLICY:

We appreciate that your raw individual EEG data has been deposited on your university supported server. However, out data deposition policy requires deposition on a static, public, site or inclusion of the data files as supplemental data. Specifically:

1) If you use Supplementary files (e.g., excel): Please ensure that all data files are uploaded as 'Supporting Information' and are invariably referred to (in the manuscript, figure legends, and the Description field when uploading your files) using the following format verbatim: S1 Data, S2 Data, etc. Multiple panels of a single or even several figures can be included as multiple sheets in one excel file that is saved using exactly the following convention: S1_Data.xlsx (using an underscore).

2) If you use data deposition: Where possible, deposition needs to be on a static repository (e.g. Zenodo, FigShare, OSF) that is publicly available, and NOT university sites or modifiable personal sites (e.g. GitHub). Once data has been deposited, please also provide the accession code or a reviewer link upon resubmission so that we may view your data before publication. 

Regardless of the method selected, please ensure that you provide the individual numerical values that underlie the summary data displayed in your figure panels. These are essential for readers to assess your analysis and to reproduce it. So we ask that you provide this data as separate files, or separate excel pages, that will allow a reader to reproduce individual figures. This should be done for the following data figures:

Fig 1C-J; Fig 2B; Fig 3B-D; Fig 4A&C; Fig 5B; Fig 6A,B,D&E, Fig 7A-J, Fig 8C-:L; Fig10

*NOTE: the numerical data provided should include all replicates AND the way in which the plotted mean and errors were derived (it should not present only the mean/average values).

*NOTE 2 (And often forgotten!!!): Please also ensure that each of the figure legends in your manuscript include a statement on where the underlying data in that figure can be found. Please ensure that your Data Statement in the submission system also accurately describes where your data can be found.

Please also ensure that any supplemental data file/s has a legend.

DATA NOT SHOWN?

Reviewer remarks:

Reviewer's Responses to Questions

PLOS authors have the option to publish the peer review history of their article (what does this mean?). If published, this will include your full peer review and any attached files.

Reviewer #1: No

Reviewer #1: The authors have sufficiently addressed my concerns.

---

## [Editor Report · Decision Letter 3]

14 Jun 2022

Dear Dr Martin,

Thank you for the submission of your revised Research Article "Neural dynamics differentially encode phrases and sentences during spoken language comprehension" for publication in PLOS Biology. On behalf of my colleagues and the Academic Editor, David Poeppel, I am pleased to say that we can in principle accept your manuscript for publication, provided you address any remaining formatting and reporting issues. These will be detailed in an email you should receive within 2-3 business days from our colleagues in the journal operations team; no action is required from you until then. Please note that we will not be able to formally accept your manuscript and schedule it for publication until you have completed any requested changes.

PRESS

Sincerely, 

Kris

Kris Dickson, Ph.D. (she/her)

Neurosciences Senior Editor/Section Manager

PLOS Biology

kdickson@plos.org